# Electroinitiated interfacial healing for external pressure-free solid-state sodium metal batteries

Tingzhou Yang[1,2,3], Siqi Qin[1,3], Shihui Gao[1], Xiaoen Wang [1] ✉, Dan Luo[1], Yu Shi [2], Qianyi Ma[2], Xinyu Zhang[1,3], Yongguang Zhang[1] ✉ & Zhongwei Chen [1,2] ✉

Solid-state sodium metal batteries with inorganic electrolytes have long been heralded as candidates for post-lithium-ion batteries. However, challenges including interfacial instability and air sensitivity continue to impede their path to commercialization. Here, we propose an interfacial healing strategy for solid-state sodium metal batteries by utilizing an electroinitiated accelerated polymerization process facilitated by charged microdroplets to increase the polymerization rate by 21.4 times. We show that the charge-driven electro-wetting enables efficient coating layers at interfaces, which impart prolonged air stability and preferentially fill voids and cracks, further constructing stable interfaces with improved compatibility and preventing dendrite-induced crack propagation. A higher critical current density of 6.8 mA cm$^{-2}$ is achieved, and assembled cells exhibit prolonged cycling life at 1.0 C over 1000 cycles. In particular, the electroinitiated accelerated polymerization-assisted interfacial healing strategy enables Ah-level pouch cells to undergo stable long-term cycling without any clamping force, demonstrating the capabilities of solid-state batteries in practical applications.

As one of the most realistic advances in energy storage, rechargeable solid-state batteries (SSBs) with metallic negative electrodes and inorganic solid-state electrolytes have promising potential to achieve high energy density and improved safety[1–4]. Such inorganic electrolytes can possess high ionic conductivities comparable to conventional liquid electrolytes (CLEs) and transference numbers close to unity, which allows fast charging capabilities without compromising battery polarization[5,6]. However, the transition from liquid-solid interfaces to solid-solid interfaces poses significant challenges, where issues related to unfavorable mechanical problems, poor metal electrode wettability, interfacial instability, and physical contact barriers become critical constraints[7–9]. Despite notable advancements, the risk of negative electrode-side dendrite propagation remains non-negligible, posing a persistent threat to interfacial stability, internal short circuiting, and

catastrophic battery failure. Unlike CLEs that can wet electrode surfaces to maintain intimate contact, it is difficult to establish stable interparticle contact between electrodes and electrolytes for SSBs[10,11]. Engineering perspectives use high-stack pressures of more than 250 MPa to improve the interfacial physical contact and apply relatively low current densities[12,13]. By doing so, many issues of SSBs can be avoided, but the additional stack pressure may be difficult or costly to achieve, further sacrificing battery practicality.

Higher stack pressure is not always better[14]. The prepared inorganic electrolytes are brittle, especially for oxide solid-state electrolytes (OSEs), and their shaping requires high pressure followed by a high-temperature annealing process, which is bound to generate microcracks on the surface, further affecting the interface stability[14,15]. Metal penetration into the ceramic drives the propagation of the

[1]State Key Laboratory of Catalysis, Dalian Institute of Chemical Physics, Chinese Academy of Sciences, Dalian, China. [2]Waterloo Institute for Nanotechnology, Department of Chemical Engineering, University of Waterloo, 200 University Ave. W., Waterloo, ON, Canada. [3]School of Chemistry and Chemical Engineering, Nantong University, Nantong, China. ✉e-mail: xiaoenw@dicp.ac.cn; ygzhang@dicp.ac.cn; zwchen@dicp.ac.cn

spallation microcracks, resulting in inhomogeneous deposition and eventual battery failure, especially under high stack pressures[16]. Furthermore, physical contact and wettability are closely relevant to the performance of SSBs[8,17,18]. The ionic diffusion between the electrode and electrolyte relies on point contacts of solid particles that are sensitive to the internal pressure of cells during cycling[19]. These issues may be exacerbated during long-term cycling, where continuous volume changes and induced stresses after using metal electrodes can aggravate crack propagation and interfacial delamination[20,21]. The accumulation of passivation products on the surface of OSEs further diminishes wettability and suppresses ionic conductivity at the metal interface, thereby enlarging the space charge layer, hindering ion transport, and increasing the interfacial resistance[22]. Therefore, understanding deposition behavior and interface engineering in OSEs is critical for promoting the practical application of SSBs[23–25]. It is imperative to develop effective and simple strategies to alleviate these problems and commercialize SSBs, especially in terms of interface engineering.

Herein, we introduce the concept of an electroinitiated accelerated polymerization (EAP) process based on charged interfacial mending glue (IMG) microdroplets to address the challenges of commercializing solid-state sodium (Na) metal batteries (SSNMBs) using OSEs operating without additional stack pressure. The EAP healing process facilitated by charged microdroplets enables prolonged air stability of OSE and Na metal, and the accompanying electrowetting interface coating effect makes it easier to cover the entire interface, where the innate voids and cracks are preferentially healed, thereby constructing the intimate interface contact with improved mechanical compatibility, further preventing the dendrite-induced crack propagation. The EAP strategy can boost Na kinetics with a higher critical current density (CCD) value of $6.8\ \text{mA cm}^{-2}$ and improve the

performance of SSNMBs pairing with various positive electrodes. The assembled 1.0 Ah solid-state pouch cell demonstrates stable electrochemical performance without adding any clamping force, paving the way for scale-up production of SSBs based on inorganic electrolytes.

## Results

### Electrowetting microdroplet coating effect

Instead of conventional anionic polymerization, the interface and practical application challenges of SSBs are addressed by filling cracks and voids via EAP strategy under an applied electric field (Fig. 1a, b, Supplementary Figs. 1–8, and Supplementary Data 1), where the charged IMG microdroplets based on ethyl 2-cyanoacrylate (ECA) monomers generated by electrospray enhance the thermodynamics of reactive chemistry and are endowed with electrowetting properties to preferentially repair defects at the interfaces between $Na_{3.4}Zr_{1.9}Zn_{0.1}Si_{2.2}P_{0.8}O_{12}$ (NZZSPO, Supplementary Figs. 9–11 and Table 1) electrolytes and electrodes, further establishing prolonged air stability of both electrolyte and Na metal, leading to uniform deposition behavior in SSNMBs and preventing the propagation of cracks inside inorganic electrolytes.

For the conventional drip-coating method, the morphology and wettability of IMG are determined by the interfacial tension when the cohesion of IMG and its adhesion to NZZSPO reach a balance (Fig. 1c). When no voltage is applied, an electric double layer is formed at the interface due to the accumulation of equal amounts of heterogeneous charges caused by the ionization, adsorption, or ion exchange. Corresponding finite element simulation (FEM) result exhibits the dynamic contact process between IMG and NZZSPO, in which the contact angle of IMG eventually returns to the actual wettability (Supplementary Fig. 12). Electrowetting refers to the optimization of the contact angle of IMG droplets on OSE by regulating the interfacial

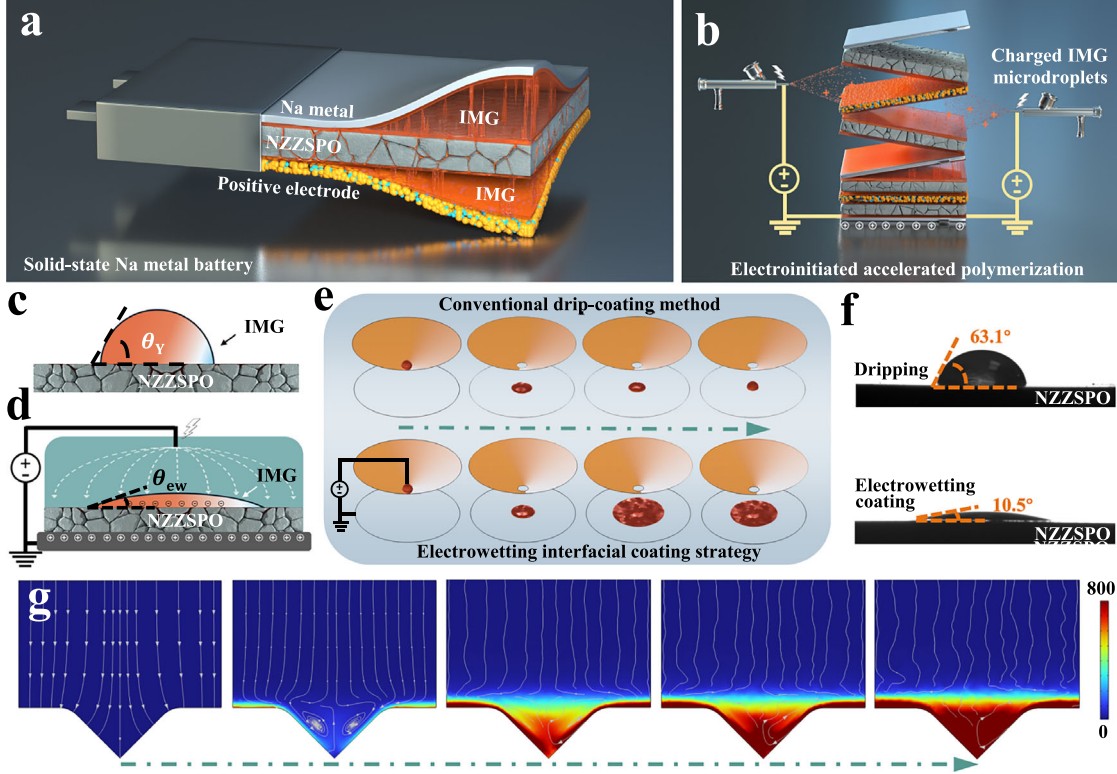

**Fig. 1 | Electrowetting interfacial coating.** Schematic diagram of (**a**) laminated structure and (**b**) assembly process of SSNMBs, where a stable interface is constructed. Sketch of IMG microdroplets sitting on NZZSPO (**c**) without and (**d**) with an applied electric field, where $\theta_Y$ and $\theta_{ew}$ are the contact angle without an applied electric field and electrowetting contact angle, respectively. **e** Simulated coating process of IMG microdroplets through conventional drip-coating method (top) and electroinitiated strategy (bottom). **f** Contact angles between NZZSPO and IMG microdroplets without (top) and with (bottom) an applied electric field. **g** Simulated flow process of EAP strategy through electrostatic spraying with priority to fill the gaps and cracks.

tension with an applied electric field according to Young-Lippmann's equation[26]:

$$\cos\theta_{ew} = \cos\theta_Y + \frac{c_H\left(V - V_{pzc}\right)^2}{2\gamma_{lg}} \qquad (1)$$

where $\theta_{ew}$ is the electrowetting contact angle, $\theta_Y$ is the contact angle without applied electric field, $V$ is the applied voltage, $V_{pzc}$ is the zero charge potential, $c_H$ is the electrical double layer capacitance per unit area, and $\gamma_{lg}$ is the interfacial tension. In our EAP strategy, IMG evenly covers the entire interface in the form of microdroplets with a net charge by applying a high electric field during the electrospray process (Supplementary Fig. 13), which exhibits an improved superhydrophilic behavior (Fig. 1d). FEM result shows that IMG microdroplet is almost spread flat on the surface of NZZSPO (Fig. 1e and Supplementary Movie 1), and can maintain the flat state without recovering to a droplet configuration due to the optimization of interfacial tension under a high electric field, which is conducive to the subsequent interface healing. The charge density of the electric double layer is increased under the high electric field, and the energy required for the IMG microdroplet to spread is reduced due to the repulsive interaction of like charges, which further leads to a decrease in the interfacial tension with IMG microdroplets, thus increasing the contact area. Such similar behavior can be observed from the contact angle test using NZZSPO and Na foil as substrates (Supplementary Fig. 14), where the EAP strategy exhibits an improved super wetting ability to NZZSPO (10.5°) compared to the conventional method without any applied electric field (63.1°, Fig. 1f). Moreover, EAP strategy can induce IMG microdroplets to penetrate completely into the cracks, voids, and roughness features of NZZSPO surface, and construct ionic conductive channels after anionic polymerization by electrons transferred from electrodes. FEM was further performed to simulate the surface concentration distribution of IMG during the electrowetting interface coating process on the NZZSPO substrate with a crack. IMG preferentially flows in from both sides of the crack until it fills the crack, and then covers the surface of the NZZSPO substrate (Fig. 1g and Supplementary Movie 2), further playing a preferential role in repairing inherent defects of OSEs.

## Electroinitiated accelerated interfacial healing

During the electrowetting interface coating process, microdroplets of IMG with a net charge alter the thermodynamics of reactive chemistry more significantly (Fig. 2a)[27,28]. When in contact with the electrode, the electron transfer process becomes thermodynamically favorable in charged IMG microdroplets obtained from the electrospray process, where electrons are transferred to the electrophilic ECA monomers and generate carbanions, which further react with the successive monomers to polymerize via a chain growth mechanism. Such an electroinitiated accelerated anionic polymerization process under the applied electric field constructs unique surface-to-surface contacts between the electrolytes and electrodes by filling the cracks and voids, thereby resolving the interface challenges of SSBs. The electroinitiated accelerated polymerization mechanism is further confirmed by the Fourier-transform infrared (FT-IR) spectrum in a glove box with relative humidity of less than 0.01 ppm to eliminate the influence of moisture on interfacial healing. The stretching vibration of C=C located at $1633\,cm^{-1}$ and bending vibrations of $=CH_2$ located at $3130\,cm^{-1}$ disappear after polymerization (Supplementary Fig. 15), confirming the successful polymerization of electrophilic monomers under the applied electric field. As shown in Fig. 2b and Supplementary Movie 3, the accelerated interfacial healing process is observed in a sealed optical cell, where the IMG microdroplets fill the cracks and gradually polymerize, resulting in improved interface physical contact. As further confirmed by the in situ FT-IR and Raman spectroscopies using cells assembled in a drying room with a dew point temperature

of less than $-35\,°C$ (Fig. 2c–f), electrons can attack the β-unsaturated carbon in ECA, forming a carbon anion, which further attacks other monomers and induces the continuous rapid polymerization reaction into long and strong chains within 15 min[29,30]. Based on the Arrhenius equation and Debye-Hückel theory[31–33], the reaction rate ($r$) of the proposed EAP strategy can be expressed as:

$$r = k_0 \cdot e^{\beta E} \cdot \left(\exp\left(-\frac{Az_i^2\sqrt{I}}{1 + Ba\sqrt{I}}\right)\right)^{m+n} \cdot [M]^n \qquad (2)$$

where $k_0$ is the intrinsic rate constant, $\beta$ is a constant that reflects the sensitivity of the electric field to the reaction rate, $E$ is the applied high electric field, $I$ is the ionic strength, and $M$ is the concentration of monomers. Compared with the conventional drip-coating method that only initiates the healing process through moisture remaining in the dry room environment and the electrolyte, the high electric field applied in our EAP strategy is bound to accelerate the polymerization reaction and shorten the reaction time. According to the in situ FT-IR spectroscopies for the conventional drip-coating method (Supplementary Fig. 16), it took about 321 min for IMG to be fully polymerized in the drying room, which was 21.4 times higher than that of the EAP strategy (Fig. 2e). For the electrolyte-electrode interface, the cross-sectional scanning electron microscopy (SEM) images revealed that large gullies could be observed using CLEs between OSE and Na metal (Fig. 2g). When the interface was covered by IMG using EAP strategy, NZZSPO was tightly bound to Na metal without any gaps (Fig. 2h), which further improved the physical contact and stabilized the interfaces, resulting in good electrochemical properties of SSNMBs. EAP treatment results in an interfacial tensile-shear strength of 4.28 MPa, providing robust mechanical adhesion that firmly anchors the electrodes to the inorganic electrolyte (Supplementary Fig. 17). Furthermore, thermogravimetric analysis showed that the residual liquid content after EAP polymerization was less than 0.3 wt% (Supplementary Fig. 18), which was much lower than that of conventional drip-coating method, indicating the formation of a predominantly solid interfacial healing layer that firmly bridges the electrodes and OSE.

Temperature-dependent conductivities of OSEs optimized by the EAP healing process or adding CLE were also measured using electrochemical impedance spectroscopy (EIS) measurements. As shown in Supplementary Figs. 19 and 20, the spraying of IMG shows increased ionic conductivity from $8.3\times10^{-3}$ to $3.6\times10^{-2}\,S\,cm^{-1}$ with the increasing temperature from 25 to 100 °C, which is higher than that of CLE. Na salts with relatively high $Na^+$ fractions and strong nucleophilicity initiate the bonding between polymerized segments and anions, while Na ions are attracted around by the Coulombic forces, making them conductive. Benefiting from the high ionic conductivity after matching the NZZSPO conductor and enhanced physical contact by the EAP strategy, the $Na^+$ transference number was calculated to be 0.90, approaching unity (Supplementary Fig. 21), which is much higher than that of the conventional drip-coating method (Supplementary Fig. 22), indicating more efficient $Na^+$ transport across the interface. As shown in Supplementary Figs. 23–25, cyclic voltammetry (CV) curves show the currents between −0.5 V and 1.0 V are assigned to Na plating/stripping on the stainless-steel electrodes. No noticeable oxidation current can be observed between 1.0 V and 5.0 V, further indicating that IMG can maintain the electrochemical stabilities of OSEs, which is much higher than that of bare CLEs (4.57 V) and in agreement with corresponding linear sweep voltammetry (LSV) results.

## Maintaining prolonged air stability

Surface degradation of OSEs in the air hinders their long-term preservation in ambient air and the commercial application of SSNMBs. As shown in Fig. 3a, the X-ray diffraction (XRD) curves confirmed that the sodiophobic sodium carbonate, sodium peroxide, and sodium

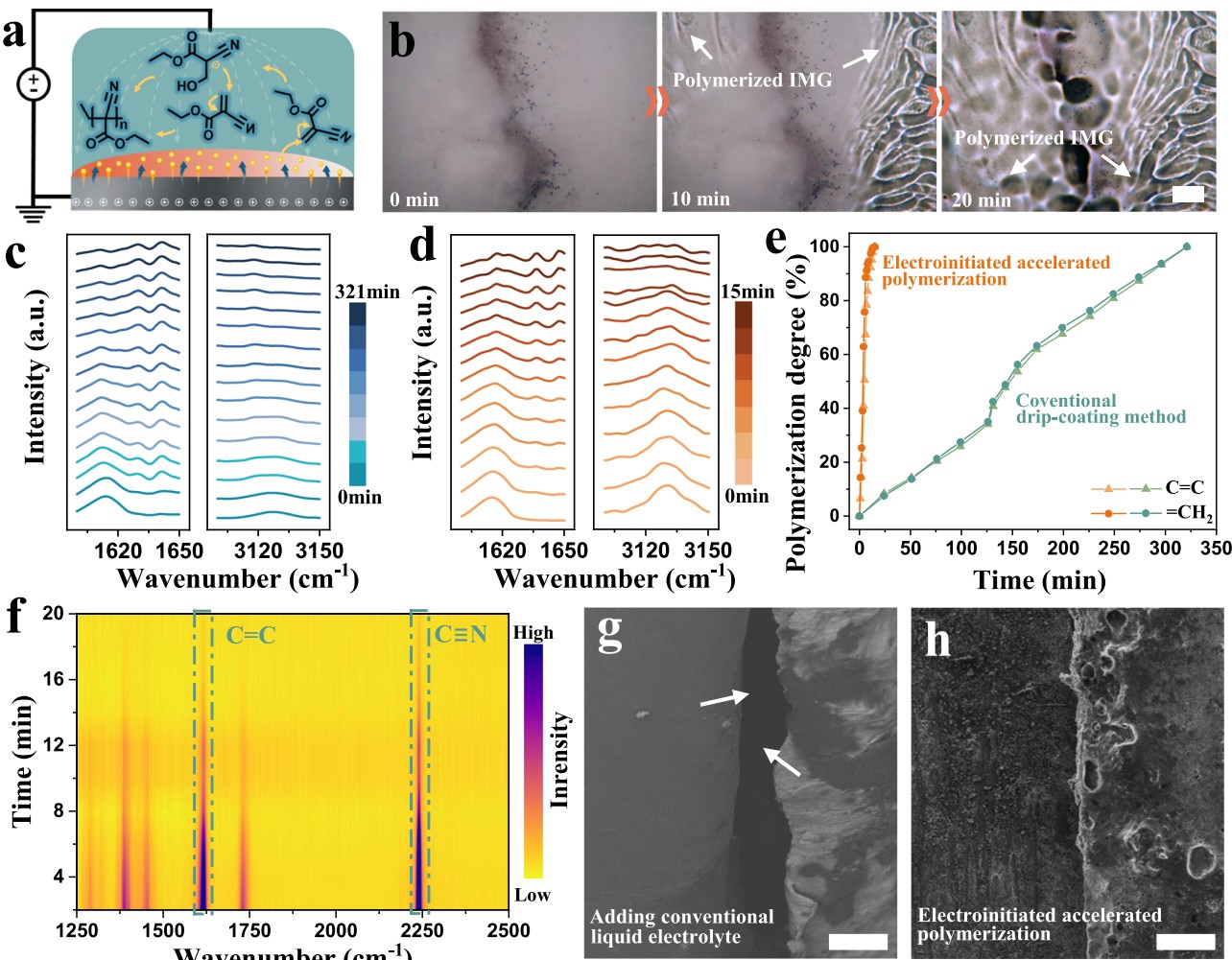

**Fig. 2 | Electroinitiatetd accelerated interfacial healing mechanism.**
**a** Accelerated healing mechanism of IMG microdrops with a net charge induced by electrons and moisture. **b** Photographs of the crack healing process using EAP strategy taken with microscopy. Scale bar, 20 μm. In situ FT-IR spectroscopy using (**c**) conventional drip-coating method with a time interval of 20 min and (**d**) EAP strategy through continuous testing. **e** Corresponding changes in the degree of polymerization over time, where the EAP strategy speeds up by 21.4 times. **f** In situ Raman of the EAP healing process in a drying room with a dew point temperature of less than − 35 °C. SEM cross-section views of SSNMB using (**g**) CLE and (**h**) EAP strategy at the interface. Scale bars, 50 μm.

hydroxides (NaOH) species were formed on the surface of NZZSPO due to the severe air corrosion caused by prolonged exposure to ambient air for more than half a year, which was not conducive to Na⁺ transfer in OSEs. EAP strategy is proposed to build an air-stable protective layer to maintain Na metal and NZZSPO stability in ambient air, which is achieved by the electroinitiated accelerated polymerization after uniform electrowetting interface spraying of IMG microdrops. No sodiophobic layer appeared on the surface after exposure to the air for half a year, further verifying the air stability of our strategy. This also applies to sulfide solid electrolytes ($Na_{2.9}PS_{3.9}Cl_{0.1}$, NaPSCl) that are unstable in air, making them air-stable (Supplementary Fig. 26 and Table 2). In addition, metallic Na is not stable in ambient air due to its high chemical reactivity, where it continuously reacts with oxygen, nitrogen, and moisture to eventually produce NaOH. This is why the color of the Na disc turns from metallic to white after being exposed to air for only two hours (Fig. 3b). In sharp contrast, the Na disc optimized by the EAP healing process still maintains a metallic color, further inhibiting the deterioration of active Na ions in the ambient air. The results were further confirmed by in situ XRD measurement, where the newly generated diffraction peaks between 27° and 53° are attributed to different planes of NaOH (Fig. 3c). There is no obvious change in XRD patterns of metallic Na optimized by EAP strategy after exposure

to ambient air for 5 h (Fig. 3d), further indicating that polymerized IMG can effectively isolate metallic Na from ambient air, further confirming the imparted air stability. As an example, we further use the Na foil, half of whose area is covered by IMG, to verify the air stability of our EAP strategy by SEM images (Fig. 3e). The naked Na foil surface becomes bumpy, which is extremely unstable in ambient air. It is indeed remarkable to see that the IMG healing layer can stabilize Na metal in ambient air, free from any atmospheric corrosion.

## Uniform sodium deposition behavior
Na metal symmetric cells with NZZSPO in which both interfaces are optimized by the EAP healing process are used to evaluate the deposition behavior without any clamping force. Figure 4a shows the long-term Na plating/stripping profiles under the current densities of 0.5 mA cm⁻² with an area capacity of 0.5 mAh cm⁻². After introducing the EAP strategy, the symmetric cell operated for more than 700 h with a stable and low overpotential of - 0.12 V. The conventional reliance of ionic diffusion on the physical contact of solid particles in SSNMBs is thoroughly optimized, where the residual gaps and cracks are filled with the conductive healing layer, further resulting in a uniform ion flux with low overpotential. A similar stable voltage plateau at steady-state can be observed when operated under 1.0 mA cm⁻²

(Supplementary Fig. 27). In contrast, the cell with CLE shows a high overpotential of up to 0.52 V, and the dendrite-induced short circuit occurs after 68 h. Many obsolete strategies show that OSE-based cells can only be operated at low depths of 0.1–0.5 mAh cm$^{-2}$ with the help of high-stack pressure, and it is critical to develop deeply cyclable SSNMBs without any stack pressures. The proposed EAP strategy can completely solve interface physical contact issues and improve the long-term stability at the interfaces, which remain stable at higher current densities between 1.5 and 3.0 mA cm$^{-2}$ with larger area capacities of 1.5 and 3.0 mAh cm$^{-2}$ in coin cells (Fig. 4b). As confirmed by EIS measurements, the overall resistance (158.7 Ω cm$^{-2}$) of the cell optimized by IMG is much lower than that of CLE (2590.9 Ω cm$^{-2}$) and conventional drip-coating method (218.8 Ω cm$^{-2}$, Supplementary Fig. 28), further demonstrating the successful elimination of interfacial issues between Na metal and NZZSPO electrolyte. According to the industry standards from the Society of Automotive Engineers (USA, SAE J2929-2011) and Recommended Chinese Automobile Standards (China, GB/T-31467.2-2015 and NB/T-33024-2016), a high working current (> 4.0 mA cm$^{-2}$) is usually required, where the solid-state symmetric cell optimized by EAP healing process exhibits a high CCD value of over 6.8 mA cm$^{-2}$ while using CLE is only 0.8 mA cm$^{-2}$ (Fig. 4c and Supplementary Fig. 29), meeting the listing requirements. Furthermore, the symmetric cell demonstrates stable rate performance from 1.0 to 6.0 mA cm$^{-2}$, characterized by a moderate increase in overpotential without noticeable fluctuations (Supplementary Fig. 30).

The open circuit energy band diagram was further investigated to clarify interfacial phenomena at the electrode-electrolyte interface (Fig. 4d and Supplementary Fig. 31). The open circuit voltage of SSNMB paired with Na$_3$V$_2$(PO$_4$)$_3$ (NVP) electrode is about 2.54 V (Supplementary Figs. 32 and 33). NZZSPO electrolyte starts reducing at −0.31 V and oxidizing at 3.52 V (Supplementary Fig. 34), which leaves a relatively high potential window for thermodynamically stable operation[34], inhibiting further decomposition of NZZSPO. Regarded as the molecular orbital theory, the components in IMG show the characteristics of being more easily oxidized and reduced due to the lower lowest unoccupied molecular orbital and higher highest occupied molecular orbital energy (Supplementary Figs. 35–37), further suggesting that our IMG is preferentially reduced on Na metal compared with NZZSPO and participates in the formation of stable solid-electrolyte interphase (SEI) layers. The role of constructing a stable interface in suppressing Na dendrite growth was further validated by SEM images (Fig. 4e), where an inhomogeneous deposition behavior can be observed after adding CLE. In contrast, the surface of Na metal optimized by polymerized IMG remains smooth and intact, confirming that the EAP strategy can construct a stable interface and induce uniform deposition in SSNMBs. The polymerized IMG formed via the EAP strategy exhibits a high Young's modulus of 7.29 GPa (Supplementary Fig. 38), significantly surpassing that of the SEI layer after adding CLE (0.74 GPa). Such enhanced mechanical rigidity provides the interfacial healing layer with a superior ability to resist deformation and effectively suppress Na dendrite penetration. Time-of-flight secondary ion mass spectrometry (TOF-SIMS) depth profiles of various interested fragments were further investigated to confirm the composition of SEI layer after cycles (Fig. 4f). Compared with adding CLEs, CN$^-$, C$_2$N$^-$, CH$_3$OCO$_2^-$, and NaF$_2^-$ fragments can be observed in the corresponding three-dimensional (3D) reconstructed images, confirming that ECA monomers join the formation of SEI layers. Similarly, extra 3D reconstructed images and corresponding normalized depth profiling further confirmed the generation and buildup of organic species, together with the continuous accumulation of inorganic interfacial species during the cycles (Supplementary Figs. 39–41). Such results were further confirmed by X-ray photoelectron spectroscopy (XPS) spectra at different etching times (Fig. 4g, h and Supplementary Figs. 42, 43). New peaks corresponding to polyester carbonyl (poly(CO$_2$)) in O 1s and C≡N in C 1s can be observed, attributing to the easier reduction of ECA

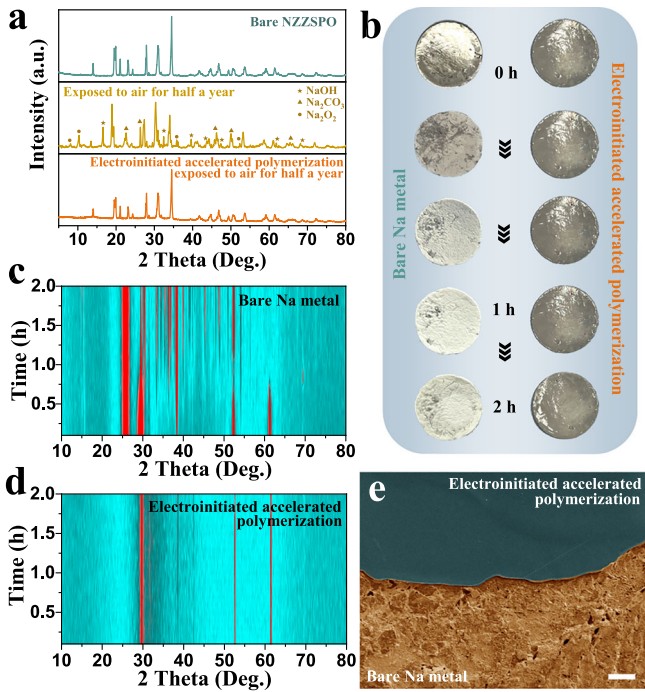

**Fig. 3 | Prolonged air stability characterization. a** XRD curves of various NZZSPO after exposure to ambient air for half a year. **b** Photographs of Na disc with and without polymerized IMG exposed to air at various times. Color plots of in situ XRD patterns of (**c**) bare Na metal and (**d**) Na metal optimized by EAP strategy in air for 2.0 h. **e** SEM image of Na metal negative electrode after exposure to the air for 2.0 h, where half of the area is covered by polymerized IMG. Scale bar, 100 μm.

monomers[35]. Such a stable SEI layer can optimize Na deposition behaviors and stabilize the interface, thereby alleviating the issues of crack propagation and interfacial delamination, leading to a longer lifespan.

## Suppressing dendrite propagation and eliminating cracks

Due to the limitations of the tableting process, some gaps and microcracks inevitably exist on the surface of OSEs, which induce the growth of Na filaments into the OSE and create stresses in the electrolyte, further lengthening and widening during repeated plating/stripping processes[14]. The continuously deposited Na metal fills the pores and squeezes the electrolyte, leading to localized crack propagation and thus shortening its lifetime. Furthermore, ionic diffusion is highly dependent on point-to-point physical contact, which is sensitive to the stress generated during Na plating/stripping processes, further resulting in crack formation and propagation. Finally, a dendrite-induced short circuit occurs when Na filaments encounter another side of the electrode (Fig. 5a). In sharp contrast, the incorporation of polymerized healing layer via EAP strategy will easily fill the gaps and microcracks through the concomitant electrowetting microdroplet coating effect while establishing stable interface contact by accelerating the electroinitiated polymerization processes, suppressing the decomposition reactions of NZZSPO and balancing the stress within SSNMBs. A uniform Na metal deposition behavior can be observed, free from any crack expansion (Fig. 5b). The morphology of cycled NZZSPO was investigated by SEM images in Fig. 5c, where both Na filaments and crack propagation can be clearly observed after adding CLEs. NZZSPO electrolyte optimized by the EAP healing process shows a smooth surface morphology without any Na filaments and cracks. To further illustrate the interfacial stability of different strategies, in situ optical microscopy observation was used to record Na metal deposition processes of OSEs optimized by different strategies (Fig. 5d). As

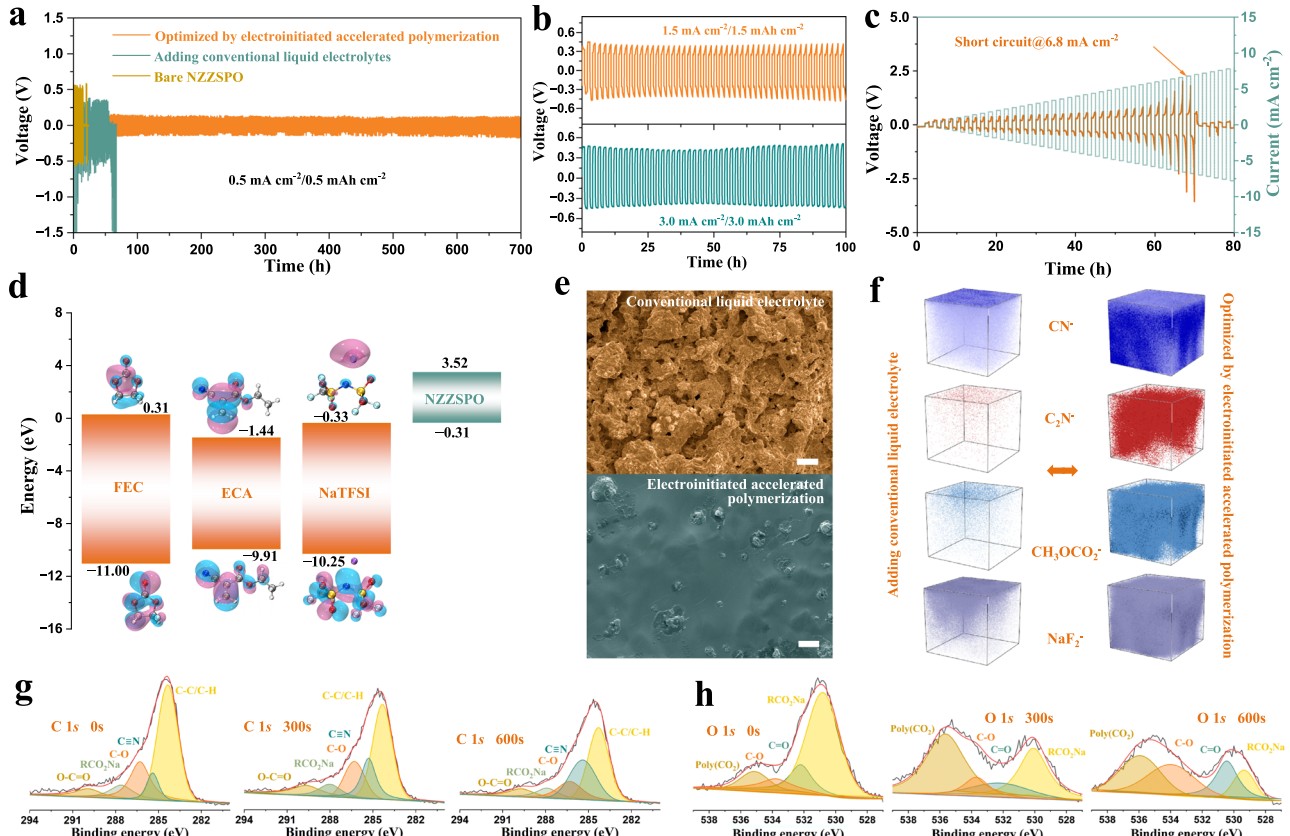

**Fig. 4 | Stable interface with uniform deposition behavior.** Voltage-time profiles of the Na metal-based solid-state symmetric cells at the current densities of (**a**) 0.5, (**b**) 1.5, or 3.0 mA cm⁻². **c** Voltage profile of symmetric cells optimized by EAP strategy at step-increase current densities. **d** Electron density distributions of various components, where the Na, S, N, C, O, and F atoms are represented by purple, yellow, blue, gray, red, and light blue spheres, respectively. **e** SEM images and (**f**) 3D reconstruction TOF-SIMS images of selected species on the surface of Na metal after 30 cycles under the current density of 0.5 mA cm⁻² at 25 °C optimized by adding CLE and EAP strategy. Scale bars, 25 µm. XPS spectra of (**g**) C 1*s* and (**h**) O 1*s* for cycled Na metal under the current density of 0.5 mA cm⁻² at 25 °C after different etching times.

time goes on, protrusions start to appear along the edge of Na metal and squeeze the internal structure of OSEs, which are unable to tolerate the large volume variations during Na plating/stripping processes, resulting in structural collapse and cracking inside OSE (Supplementary Movie 4). In contrast, the edge of symmetric cells optimized by the EAP strategy remains smooth, and a uniform deposition process can be observed without any Na dendrites or protrusions (Supplementary Movie 5). We further use FIB-SEM and nanoscale X-ray computed tomography (Nano-CT) to track the crack initiation and propagation during the plating process (Supplementary Fig. 44). On plating, Na deposits occur throughout the pores and microcrack surfaces, progressively filling them. These uneven depositions can cause pressure buildup accompanied by crack propagation, in which Na fills the cracks (Fig. 5e). After introducing an electroinitiated healing layer, the NZZSPO remains intact without any cracks and dendrite structure. Similar phenomena can also be observed in 3D reconstruction Nano-CT images. After cycling, the addition of CLEs failed to establish a stable and robust interfacial layer, where long bifurcated cracks can be found inside NZZSPO (Fig. 5f). Na protrusions grow through the NZZSPO electrolyte, eventually short-circuiting SSNMBs. In contrast, a uniform plating/stripping process was obtained after introducing the EAP strategy, further avoiding the crack expansion caused by substantial internal stresses.

## Performance of solid-state batteries

For conventional SSNMBs, the discharging capacity relies on external stack pressure to improve the interface contact. However, in addition to increasing costs, introducing higher stack pressure will lead to the expansion of cracks and accelerate failure, which is not conducive to large-scale commercial applications. For the EAP strategy, all SSNMBs were assembled in the coin cells or pouch cells without applying any clamping force (Supplementary Figs. 45 and 46). A stable interface is constructed, which prevents the dendrite initiation and propagation, ensuring that the SSB operates efficiently without external stack pressure. As shown in Fig. 6a, the coin cell with bare NZZSPO electrolyte and Na₃V₂(PO₄)₃ (NVP) electrode failed directly and could not release any capacity due to poor physical contact. Adding CLEs could improve partial physical contact, but the capacity deteriorated severely after 82 cycles due to the propagation of the cracks and growth of Na dendrites. Compared with CLEs, introducing the EAP healing process can significantly reduce the resistance of SSNMBs (Supplementary Fig. 47), which exhibit the highest capacity retention for more than 1000 cycles with an average Coulombic efficiency of 99.97%. Figure 6b and Supplementary Fig. 48 show the cycling performance of SSNMBs optimized by EAP strategy with different active material loading from 3.1 and 7.2 to12.5 mg cm⁻², which deliver area capacities of 0.34 and 0.66 to 1.10 mAh cm⁻² at 0.5 C (1.0 C = 127.0 mA g⁻¹). Even with an active material loading of 26.3 mg cm⁻² (Fig. 6c), the capacity still maintains 2.16 mAh cm⁻² with a capacity retention of 98.00% and Coulombic efficiency of 99.95% after 100 cycles at 0.2 C. Adding CLE proves ineffective at elevated active material loading, and the conventional drip-coating method often results in unstable discharging capacities with pronounced fluctuations under high-loading conditions (Supplementary Figs. 49 and 50).

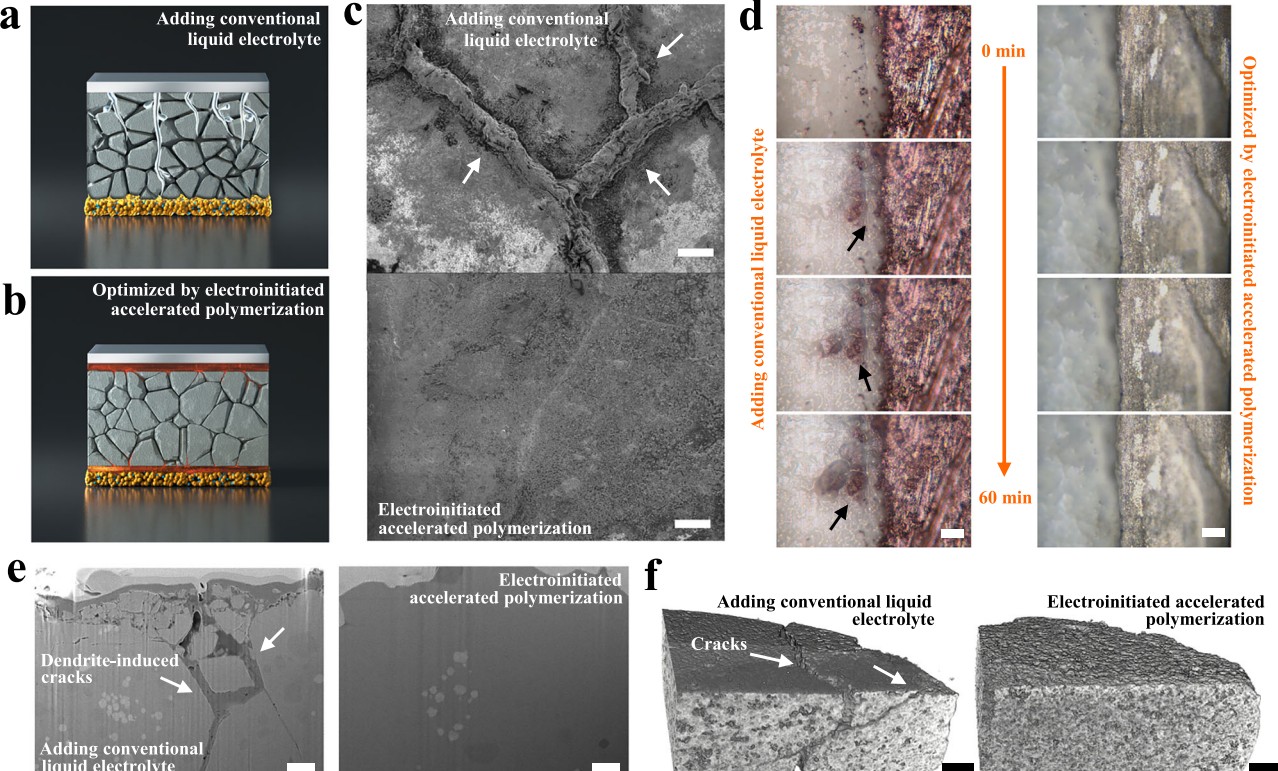

**Fig. 5 | Evaluation of dendrite propagation and crack expansion.** Illustration of dendrite and crack initiation and propagation in SSNMBs optimized by (**a**) CLE and (**b**) EAP strategy. **c** SEM images of OSEs after 20 cycles at 25 °C, where interfaces are optimized by the CLE (top) and EAP strategy (bottom) under a current density of 3.0 mA cm⁻². Scale bars, 10 μm. **d** In situ optical microscopy observations of OSEs based on different optimized strategies under a current density of 3.0 mA cm⁻² at 25 °C. Scale bars, 5 μm. **e** FIB-SEM and (**f**) Nano-CT images of OSEs after 30 cycles in symmetric cells under a current density of 2.0 mA cm⁻² at 25 °C using different interfacial optimization methods. Scale bars, 200 nm.

Figure 6d presents the rate performance, where SSNMB optimized by polymerized IMG delivers a higher capacity than that with CLEs, especially at high specific currents of 5.0–15.0 C, indicating that introducing the EAP strategy can optimize physical contact and accelerate charge transfer at the electrode/electrolyte interface, endowing the battery with improved fast charging capabilities. By contrast, the SSNMB with CLEs exhibits a serious capacity decay with gradually increasing current density, and no capacity can be delivered when the current exceeds 5.0 C. Furthermore, the Na|NZZSPO|NVP solid-state laminated pouch cell optimized by the EAP healing process was prepared and exhibited improved cycling performance after 250 cycles at 0.1 C with a negative/positive capacity ratio (N/P ratio) of 4.53 (Fig. 6e and Supplementary Figs. 51–53). To further verify the practicality of our electroinitiated interfacial healing engineering strategy, the coin cells with NZZSPO electrolyte and NaNi$_{0.33}$Fe$_{0.33}$Mn$_{0.33}$O$_2$ (NFM) electrode were tested in the voltage window of 2.0–4.0 V. Supplementary Fig. 54 exhibited intuitionistic particle cracks in the cycled NFM electrode for SSNMBs optimized by adding CLE, whereas the EAP strategy still maintained its integrity morphology without cracks even after 50 cycles. The specific capacities of SSNMBs optimized by the EAP strategy with active materials loading of 9.8, 13.9, 17.7, and 25.0 mg cm⁻² are shown in Fig. 6f, where the discharge capacities are stable at 1.12, 1.65, 2.16, and 3.19 mAh cm⁻², respectively, matching the levels of commercially available lithium-ion batteries. The corresponding discharging/charging curves further confirmed the fast kinetics with high reversibility (Fig. 6g). A stable cycling performance is achieved at a high current of 5.25 mA cm⁻² (Supplementary Fig. 55). To provide evidence that our EAP strategy has practical usage in industry and can enable the commercialization of SSNMBs, an 1.0 Ah Na|NZZSPO|NFM solid-state laminated pouch cell

was also tested at 0.1 C with a N/P ratio of 3.09 (Fig. 6h and Supplementary Fig. 56), which delivered a reversible capacity of 979.3 mAh with a capacity retention of 96.25% and Coulombic efficiency of 99.92% after 100 cycles. Similarly, the EAP strategy is also applicable to other solid-state conductors, among which NaPSCl-based sulfide solid electrolytes show significantly improved performance in symmetric cells and SSNMBs with NFM electrodes (Supplementary Figs. 57 and 58), at the forefront of state-of-the-art SSNMBs (Supplementary Figs. 59, 60, and Table 3).

## Discussion

In summary, an electroinitiated interfacial healing strategy is proposed to enable stable cycling in practical air-stable SSNMBs that operate well without external stack pressure. A complanate interface layer with enhanced compatibility for efficient ionic migration is obtained after introducing the EAP healing process facilitated by charged microdroplets, which is 21.4 times faster than conventional methods. The congenital defects inside inorganic conductors are preferentially filled through the electrowetting interface coating effect, thereby constructing a stable interface and alleviating dendrite propagation-induced crack expansion. The interface-optimized SSNMBs paired with various positive electrodes exhibit reliable electrochemical performance under high specific currents up to 15.0 C or deliver a high area capacity of 3.19 mAh cm⁻², and Ah-level laminated solid-state Na metal pouch cells are also presented without applying any clamping force. We believe that this work opens a path to exploit electroinitiated wettability and interface optimization to address challenging interfacial dynamics and provides valuable insights for developing practical SSBs with various inorganic electrolytes.

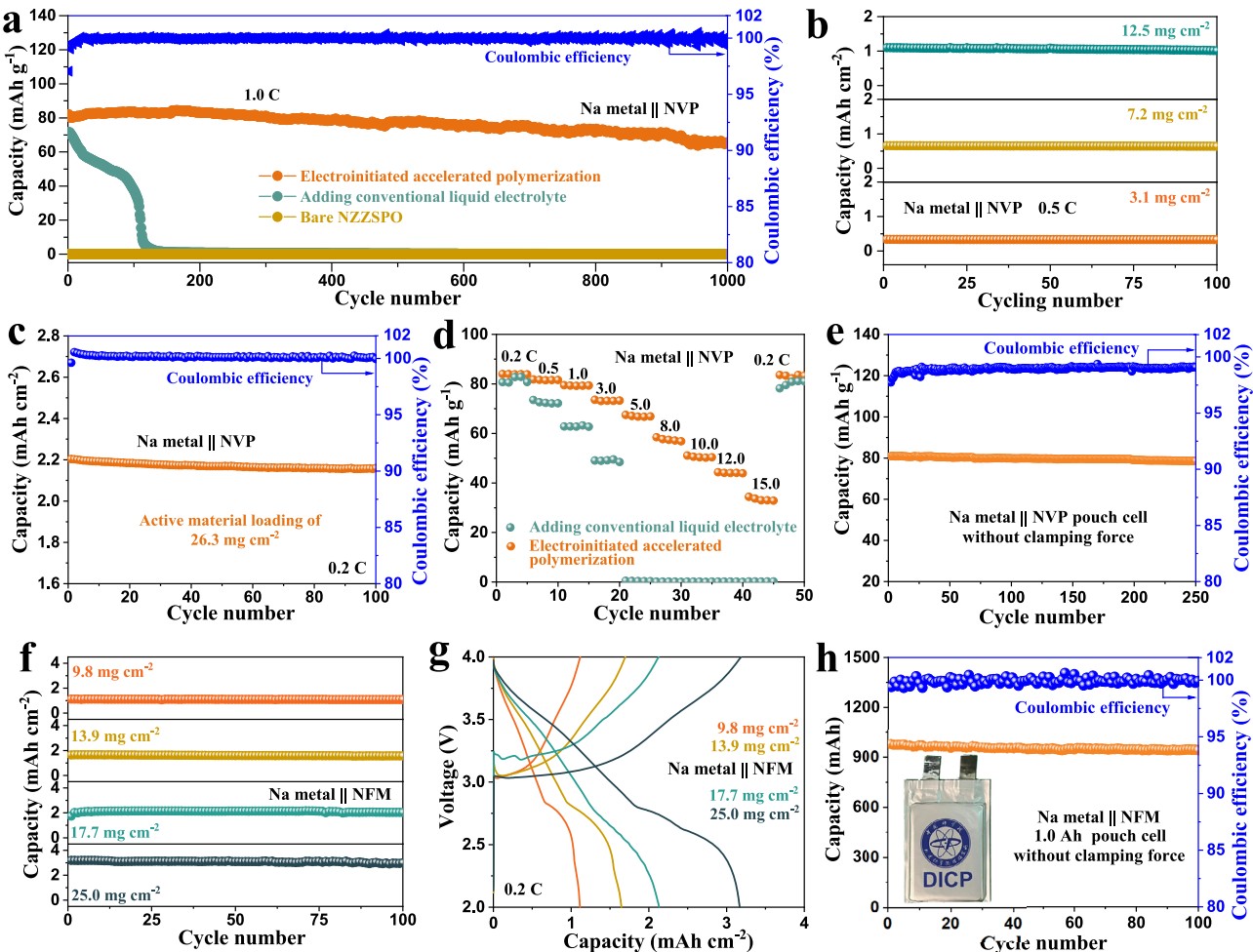

**Fig. 6 | Properties of SSNMBs without external stack pressure. a** Long-term cycling stability of SSNMBs paired with NVP electrodes at 1.0 C. **b** Cycling performance of SSNMBs optimized by EAP strategy with different active material loadings at 0.5 C. **c** Cycling stability with high loading NVP electrode about 26.3 mg cm⁻² at 0.2 C. **d** High rate performance of SSNMBs using different interfacial optimization methods. **e** Cycling performance of Na|NZZSPO | NVP solid-state laminated pouch cell at 0.1 C optimized by EAP strategy without any clamping force. **f** Cycling performance and (**g**) corresponding discharging/charging curves of SSNMBs paired with different active material loadings of NFM electrodes at 0.2 C. **h** Cycling performance of 1.0 Ah Na|NZZSPO | NFM solid-state laminated pouch cell at 0.1 C optimized by EAP without any clamping force.

## Methods

### Materials

ECA (97%), sodium block (99%), sodium carbonate (Na₂CO₃, 99.5%), zirconium dioxide (ZrO₂, 99%), zinc oxide (ZnO, 99.99%), silicon dioxide (SiO₂, 99.8%), ammonium dihydrogen phosphate (NH₄H₂PO₄, 99.99%), ammonium vanadate (NH₄VO₃, 99.95%), and citric acid (99.5%) were purchased from Sigma-Aldrich. Fluoroethylene carbonate (FEC, 99.9%), ethylene carbonate (EC, 99.95%), dimethyl carbonate (DMC, 99.99%), sodium hexafluorophosphate (NaPF₆, 99.8%), sodium bis(trifluoromethylsulfonyl)imide (NaTFSI, 99.5%), and sodium perchlorate (NaClO₄, 99.9%) were obtained from Dodochem and used as received. Na discs with a thickness of 0.45 mm were bought from Guangdong Canrd New Energy Technology. NaNi₀.₃₃Fe₀.₃₃Mn₀.₃₃O₂ powder was obtained from Zhejiang Natrium Energy.

### Preparation of OSEs

A solid-state reaction was used to synthesize the NZZSPO powder by mixing stoichiometric amounts of Na₂CO₃, ZrO₂, ZnO, SiO₂, and NH₄H₂PO₄ with the additional 15 wt% Na₂CO₃ and NH₄H₂PO₄ in the planetary ball milling (Restch Pulverisette 5) for 4.0 h. 5–10 mL of absolute alcohol is added to achieve a homogeneous state. The mixture was then dried at 50 °C and calcinated at 700 °C in the air for 10 h.

After grinding by the planetary ball milling, the obtained powders were screened through a 200 mesh sieve and mixed with 10 wt% PVA binder. The obtained mixture was isostatic pressed into pellets (200 MPa, diameter of 12 mm) and preheated at 650 °C for 2 h to remove the binder. The NZZSPO electrolyte was obtained by further calcinating at 1200 °C for 12 h.

### Preparation of NVP powder

NH₄VO₃ (4.86 g), NaH₂PO₄ (8.52 g), Citric acid (7.68 g), and graphene oxide (0.4 g) were dissolved in deionized water (60 mL) and vigorously stirred for 4 h to obtain a mixture solution. The precursor solutions were freeze-dried for 12 h after pre-freezing in liquid nitrogen. After reground, the mixture was annealed at 800 °C for 8 h under an Ar-H₂ (95:5 in vol.) atmosphere.

### Electrochemical characterization

The NVP electrode was made by a mixture of 90 wt% NVP powder, 5 wt% carbon black (Timcal), and 5 wt% PVDF (Solvay 5130, molecular weight of 1 million), and the NFM electrode was made by a mixture of 92 wt% NFM powder, 5 wt% Super P, and 3 wt% PVDF. Aluminum foil (16 μm, 99.6%) is used as the current collector of positive electrodes with an active material loading of 2.6–3.2 mg cm⁻². Our IMG (1.0 M NaPF₆ in

ECA:FEC with different volume ratios) and CLE (1.0 M NaClO$_4$ in EC:DMC (1:1 in vol.) with 5.0 wt% FEC) were prepared in the glove box. The thickness of the NZZSPO pellet was controlled at 125.0 μm ± 5.0 μm. The improved liquid electrostatic spraying gun (LVOBO, BO-818) was used for EAP processes. A high-voltage electric field of +15.0 kV was applied to the spraying needle, while the NZZSPO pellet substrate was connected to − 2.0 kV and mounted on a vacuum chuck to ensure effective droplet deposition and uniform coverage. The spraying distance was maintained at approximately 8–15 cm, allowing for optimal electrowetting and spreading of IMG microdroplets at the electrolyte interface. The IMG flow rate was precisely controlled at 0.06 mm min$^{-1}$, where a single side of coin cells takes 1–3 s and a single side of pouch cells takes 5–7 s. The entire process occurred during battery assembly, with IMG covering the entire NZZSPO electrolyte surface and then quickly sandwiched between the positive electrode and Na metal. As a contrast, 15–20 μL cm$^{-1}$ CLEs were evenly coated on both sides of the NZZSPO electrolyte. The optimized NZZSPO electrolytes with a diameter of 16.0 mm were sandwiched by the prepared positive electrode with a diameter of 12.0 mm and Na foil discs (0.45 mm) with a diameter of 15.6 mm to build the batteries in CR2032-type coin cells with a stainless steel case and spring (Canrd, funnel-shaped CR20, 1.1 mm). The average value of ionic conductivity and viscosity were obtained through multiple tests using Orion Star A322 Conductivity Portable Meters and HAAKE Viscotester 3 rotational viscometer (ThermoFisher Scientific). For pouch cells, the dimensions of the positive electrode, NZZSPO electrolyte with a thickness of 80.0 μm ± 5.0 μm, and Na foil are 43 mm × 5 mm, 47 mm (± 1.5 mm) × 6 mm (± 1.5 mm), and 45 mm × 5 mm, respectively. Na foil was prepared by repeated rolling, and its thickness was precisely measured using an electronic thickness gauge (DLX-chg30). 14 pieces of positive electrodes with an active material loading of 15 mg cm$^{-2}$ and 15 pieces of homemade Na foil with a thickness of 50 μm (± 5 μm) are stacked together by a manual lamination fixture (MSK-111A-PF4662, Shenzhen Kejing Star Technology) and encapsulated by aluminum plastic film through a vacuum final sealing machine (MSK-115A-L, Shenzhen Kejing Star Technology). Both sides of the NZZSPO electrolytes are coated with a layer of IMG by the improved electrostatic spraying method. CR2032-type coin cells and pouch cells activated at 25 ± 2 °C for 12 h were measured using a Neware battery test system (MIHW-200-160CH-B with a constant testing temperature of 25 °C) at a voltage window of 2.0–4.0 V without external stack pressure and precycling procedure. A thin layer of Au (thickness ~ 300 nm) was sputtered on one side of a stainless steel pellet using a sputter coater (Saintins, JS-1600) and acted as a blocking electrode. AC impedance analysis from 7 MHz to 1 Hz with an amplitude of 10 mV was used to measure the ionic conductivity (BioLogic VMP3 potentiostat) with the Au|SSE|Au symmetric cells based on the following Equation:

$$\sigma_t = \frac{L}{R_t S} \qquad (3)$$

where $S$, $R_t$, $L$, and $\sigma_t$ represent the surface area of the Au electrode, resistance value, thickness of the NZZSPO pellet, and ionic conductivity. CV curves were collected between − 0.5 and 5.5 V at a scan rate of 1.0 mV s$^{-1}$, and LSV was conducted using a Na|electrolyte| stainless steel disk at a scan rate of 0.2 mV s$^{-1}$ from 2.0 to 6.0 V (vs. Na/Na$^+$) at 25 ± 2 °C. For symmetric cells, two identical Na discs were assembled, and the galvanostatic stripping-plating processes were recorded at room temperature with various current densities for 0.5 to 3.0 mA cm$^{-2}$. The CCD was evaluated using symmetric cells with an increasing step of 0.2 mA cm$^{-2}$.

## Material characterization
The morphologies of NZZSPO electrolyte and Na metal after cycling were investigated using field-emission scanning electron microscopy

(SU8010, Hitachi), where the cycled NZZSPO electrolyte was washed by N,N-dimethylformamide (Sigma, 99.8%) or propylene carbonate (Sigma, 99.7%) solution. A rough cross-sectional milling was done on the cycled NZZSPO electrolyte by using a 30 kV Ga$^+$ beam with a current of 0.1–0.5 nA. FIB-SEM images were collected using a FEI Scios DualBeam FIB/SEM system with an Everhart-Thornley detector at 2.0 kV and 0.2 nA. In situ FT-IR spectroscopy was collected via FT-IR spectrometers (INVENIO S, Bruker) using reflection mode fitted with an in situ battery mold with ZnSe window at 25 ± 2 °C in the dry room under a dew point temperature of less than − 35 °C, and the corresponding galvanostatic stripping-plating processes were carried out by CHI660E (Shanghai Chenhua Instrument) electrochemical workstation. In situ XRD investigations were recorded by a X-ray powder diffractometer (Empyrean S3) with Cu Kα radiation under 5° min$^{-1}$ with a relative humidity of 30–45% at 25 ± 2 °C. Thermogravimetric analysis (NETZSCH STA 449F5) was collected at 5.0 °C min$^{-1}$ between 25 °C and 800 °C under nitrogen atmosphere. The components of the SEI layer on the surface of cycled Na metal were investigated by TOF. SIMS5-100 using a Cs$^+$ ion beam with the energy of 1.0 keV and current of 40 nA, and XPS analysis (ESCALAB 250Xi, Thermo Scientific) equipped with a monochromatic Al Kα X-ray source (1486.6 eV, voltage 15 kV, emission angle 58°, power 150 W, and spot size 500 μm). Before XPS and TOF-SIMS analyses, Na metal after 30 cycles was washed with propylene carbonate solution and dried in an argon-filled glovebox. An in situ battery mold fitted with highly transparent quartz glass (STC-Q, KJ Group) was obtained for optical microscopy study. The contact angle measurements were performed using a DSA25S instrument with the sessile drop method, where a 2 μL droplet was deposited onto the surface of the substrate and recorded by a high-resolution camera. $^{23}$Na NMR was performed by JNM-ECZL400S (JEOL, ceramic rotor, 400 MHz), and interfacial mending glue was placed in an NMR tube along with a sealed capillary tube containing 1.0 M NaClO$_4$ in D$_2$O as an external reference.

## Theoretical calculations
Density functional theory calculations were carried out using Gaussian 16 programs. Geometric optimizations were performed using M06-2X hybrid functional with Grimme's dispersion correction of the D3 version. The standard 6-311 + G** basis set for all atoms was used. Frequency calculations at the same level of theory have also been performed to identify all stationary points as minima (zero imaginary frequencies). Approximate solvent effects were taken into consideration based on the IEFPCM continuum solvation model in all calculations. The isosurfaces of canonical molecular orbitals are obtained by the Multiwfn and VMD programs. The electron affinity ($E_A$) and ionization potential ($I_P$) are respectively defined as:

$$E_A = E_N - E_{N+1} \qquad (4)$$

$$I_P = E_{N-1} - E_N \qquad (5)$$

where N is the number of electrons. As implemented in the Vienna Ab Initio Simulation Package, density functional theory within the projector augmented plane-wave approach was used for the open-circuit energy calculations for the SSB system. For the exchange-correlation functional, the generalized gradient approximation with the Perdew-Burke-Ernzerhof formulation was adopted. The plane-wave basis set was truncated at a cut-off energy of 480 eV, and the iterative solution of the Kohn-Sham equations was considered converged when the energy difference fell below 10$^{-5}$ eV. Atomic positions were optimized until the residual forces on all atoms were reduced to below 0.02 eV Å$^{-1}$. A vacuum region of 20 Å was introduced along the direction perpendicular to the slab to minimize any interactions between neighboring layers[36].

## Finite element simulation

The corresponding finite element simulations and analysis were designed to provide a qualitative and illustrative representation of droplet spreading and crack healing processes under electrowetting conditions. The coupled calculations were performed using the phase field and laminar flow modules for finite element analysis of the changes in the contact angle of interfacial mending glue microdroplets. The specific structural model and material parameters are exhibited in Supplementary Fig. 61. For the phase field module partial differential equations:

$$\frac{\partial \varnothing}{\partial t} + u \cdot \nabla \varnothing = \nabla \cdot \frac{\gamma \lambda}{\varepsilon_{pf}^2} \nabla \psi \tag{6}$$

$$\psi = -\nabla \cdot \varepsilon_{pf}^2 \nabla \varnothing + \left( \varnothing^2 - 1 \right) \varnothing \tag{7}$$

where $\varnothing$ is the phase variable of the phase field (air is $-1$ and droplet is 1), $u$ is the fluid velocity field, $\lambda$ is the mixing energy density, $\gamma$ is the thickness parameter of the interface, $\varepsilon_{py}$ is the capillary width (control parameter of interface thickness), and $\psi$ is the auxiliary variable of the phase field. Among them, $\lambda$ and $\gamma$ are used to control the convergence of the model:

$$\lambda = \frac{3\sqrt{2}}{4} \delta \cdot \varepsilon_{pf} \tag{8}$$

$$\gamma = x \cdot \varepsilon_{pf}^2 \tag{9}$$

where $x$ is the adjusting parameter for mobility. For the position of the phase field boundary:

$$n \cdot \frac{\gamma \lambda}{\varepsilon_{py}^2} \nabla \psi = 0 \tag{10}$$

$$n \cdot \varepsilon_{py}^2 \nabla \varnothing = \varepsilon_{py}^2 \cos(\theta_w) |\nabla \varnothing| \tag{11}$$

where $\theta_w$ is the contact angle, satisfying the dynamic contact angle equation:

$$\frac{\cos(\theta_e) - \cos(\theta_w)}{\cos(\theta_e + 1)} = \tanh\left(4.96 \cdot Ca^{0.702}\right) \tag{12}$$

where Ca is a dynamic parameter between $4*10^{-5}$ and 36. For the laminar flow module partial differential equations:

$$\rho \frac{\partial u}{\partial t} + \rho(u \cdot \nabla)u = \nabla \cdot \left[ -pI + \left( \nabla u + (\nabla u)^T \right)^T \right] + F + \rho g \tag{13}$$

$$\rho \nabla \cdot u = 0 \tag{14}$$

where $\rho$ is the material density, $u$ is the spatial fluid velocity field, $p$ is the fluid stress tensor, $I$ is the unit tensor matrix, $F$ is the applied volume force, $g$ is the gravitational acceleration, and $\mu$ is the dynamic viscosity. The droplet density is 1.41 kg m$^{-3}$ with the viscosity of 9.96 mPa s. The applied electric field is achieved by adjusting the contact angle and surface tension of the droplet with initial contact angles of 10.5° and 63.1° for the EAP strategy and conventional drip-coating method. The surface roughness of the receiving bottom is set to 48.7 nm based on the root mean square roughness of OSEs. Furthermore, a two-dimensional finite element model has been established to investigate the crack healing process at the electrode-electrolyte interface under the electric field with the Nernst-Planck and the Laminar flow interface. The simulation area has a height of 7.0 μm

and a width of 8.0 μm. The mesh is chosen to be triangular or tetrahedron-based while using an increasing refinement toward the electrode bands. The adsorption of IMG on the surface of OSE is determined by the caused by concentration changes, migration in the electric field, and convection caused by density changes, as shown in the following equations:

$$\frac{\partial_{C_i}}{\partial_t} = -\nabla \cdot J_i \tag{15}$$

$$J_i = -D_i \nabla c_i - z_i \mu_i F c_i E + c_i \vec{v} \tag{16}$$

$$\mu_i = \frac{D_i}{RT} \tag{17}$$

$$\rho \frac{\partial \vec{v}}{\partial t} + \rho(\vec{v} \cdot \nabla)\vec{v} = -\nabla p + \mu \nabla^2 \vec{v} + \rho g \tag{18}$$

$$\rho \nabla \cdot \vec{v} = 0 \tag{19}$$

where $Ci$ is the concentration, $J_i$ is the flux vector, $Z_i$ is the charge number, $\mu_i$ is the ion mobility, $F$ is the Faraday constant, $D_i$ is the diffusion coefficient, $\vec{v}$ is the velocity field, $p$ is the pressure of the electrolyte, $\rho$ is the density, and $\mu$ is the dynamic viscosity. In the simulation, the gravitational acceleration is set to 9.8 m s$^{-2}$, and the Na$^+$ diffusion coefficient is taken as $1.33 \times 10^{-9}$ m$^2$ s$^{-1}$. The Faraday constant is 96485 C mol$^{-1}$ with the charge number of 1. The dynamic viscosity of the droplet is 9.96 mPa s, while the potentials applied at the top and bottom boundaries were 15.0 kV and $-2.0$ kV, respectively.

## Reaction rate of EAP healing process

We first introduce the basic equation of the reaction rate for anionic polymerization[31], which can be expressed as:

$$r = k \cdot [M]^n \tag{20}$$

where $r$ is the reaction rate of the polymerization process, $k$ is the rate constant, $M$ is the concentration of ECA monomers, and $n$ is the reaction order. In our EAP healing process, $k$ can be affected by the electric field and the initiator concentration (moisture in the air and electrolytes). Based on the Arrhenius equation[32], applying an electric field can reduce the activation energy of the reaction, thereby accelerating the polymerization reaction. The electric field induced $k$ is

$$k_E = A e^{-\frac{E_a - \mu E}{RT}} \tag{21}$$

where $R$ is the gas constant, $T$ is the absolute temperature, $E_a$ is the activation energy, $E$ is the applied electric field, and $\mu$ is the dipole moment of the reactant or intermediate. Under the high electric field, the corresponding induced $k$ is

$$k_E = k_0 \cdot e^{\beta E} \tag{22}$$

where $k_0$ is the intrinsic rate constant, $\beta$ is a constant that reflects the sensitivity of the electric field to the reaction rate, and $E$ is the applied high electric field. According to the Debye-Hückel theory and ion effect[33], the initiator concentration-induced k is defined as:

$$k_I = k_0 \cdot \gamma^{m+n} \tag{23}$$

where $k_0$ is the intrinsic rate constant in the absence of ionic strength and $\gamma$ is the activity coefficient of the initiator (moisture in the air and

electrolytes). Affected by ionic strength ($I$), $\gamma$ can be defined as:

$$\gamma = \exp\left(-\frac{Az_i^2\sqrt{I}}{1+Ba\sqrt{I}}\right) \qquad (24)$$

where $I$ satisfies:

$$I = \frac{1}{2}\sum z_i^2[i] \qquad (25)$$

A higher moisture concentration in the dry room can increase $I$ and reduce $\gamma$, further promoting the polymerization reaction. In our EAP strategy, the abundant electrons from the electrode initiate accelerated polymerization reactions under a high electric field. Compared with moisture-induced polymerization reactions, the reaction rate for our EAP healing process can be expressed as:

$$r = k_0 \cdot k_E \cdot k_I \cdot [M]^n = k_0 \cdot e^{\beta E} \cdot \left(\exp\left(-\frac{Az_i^2\sqrt{I}}{1+Ba\sqrt{I}}\right)\right)^{m+n} \cdot [M]^n \qquad (26)$$

Under the same external environment in the dry room under a dew point temperature of less than $-35\,°C$, the proposed EAP strategy is bound to accelerate the polymerization reaction of charged IMG microdroplets.

## Data availability
All data supporting the findings of this study are available within the article and the Supplementary Information file. Source data are provided in this paper.

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

## Acknowledgements

We acknowledge the support from the Strategic Priority Research Program of the Chinese Academy of Sciences (XDB0600400, Z.C.), Liaoning Binhai Laboratory (LBLB202304, Z.C.), Dalian Revitalization Talents Program (2022RG01, Z.C.), Vacuum Interconnected Nanotech Workstation (Nano-X, Suzhou Institute of Nano-Tech and Nano-Bionics), the Waterloo Institute for Nanotechnology, the Natural Sciences and Engineering Research Council of Canada, and the Program for Jiangsu Specially-Appointed Professors. We also thank Scientific Compass (www.shiyanjia.com) for their invaluable assistance in material characterization and simulation.

## Author contributions

T.Y., X.W., Y.Z. and Z.C. planned the research and analyzed the results. T.Y. drafted the manuscript and analyzed the data. S.Q. and S.G. completed most of the experiments. D.L., Y.S., Q.M. and X.Z. supervised the project and assisted in material characterization. X.W., Y.Z. and Z.C. reviewed and edited the manuscript. All authors discussed and contributed to the results.

## Competing interests

The authors declare no competing interests.
