## [Transparent Peer Review file · Nature Communications]

Electroinitiated interfacial healing for external pressure-free solid-state sodium metal batteries

Corresponding Author: Professor Zhongwei Chen

Version 0:

Reviewer comments:

Reviewer #1

(Remarks to the Author)

This manuscript introduces an electroinitiated accelerated polymerization (EAP) strategy that utilizes charged microdroplets to facilitate interfacial healing in solid-state sodium metal batteries (SSNMBs) based on modified NASICON-based solid electrolytes. Particularly, the strategy enables improvements in critical current density (CCD) and long-term cycling stability and unlock pressure-free SSNMBs. The authors present computational analysis along with a wide range of data, including in situ spectroscopies, TOF-SIMS, XPS, FIB-SEM, Nano-CT, and pouch cell testing, in support of their claims. While the work is comprehensive, well-organized and interesting, I have concerns about the mechanistic foundation, novelty compared to existing strategies, and the lack of rigorous benchmarking performances. Unfortunately, these issues significantly limit the manuscript's suitability for publication in Nature Communications. I will elaborate my detailed comments below for the authors' consideration in further improving the manuscript.

(1) The topic is timely and addresses one of the most pressing issues in the commercialization of all-solid-state sodium batteries. However, prior work on interfacial polymer coatings or plastic crystal coatings have demonstrated similar benefits in mitigating voids and cracks in various solid electrolytes (i.e., doi/full/10.1021/acscentsci.6b00321). While the use of ECA and electroinitiated polymerization adds a fresh angle, the novelty is somewhat incremental unless mechanistic advantages are better substantiated.

(2) The absence of stacking pressure is demonstrated as a key breakthrough, nonetheless the interfacial resistance is still quite high after the modification (319 Ohm), although the area is not provided to estimate the ASR. In comparison, it is noticeable that some recent interfacial strategy such as alloying based interface (Na-Pb, Na-Sn, etc.) can achieve near zero resistance while also supporting zero stacking pressure. Considering the key findings of this work, the reviewer would suggest the authors provide a concise overview of key anode issues on current NASICON-based SSNMBs in the introduction part. Also, a comparison in key performance metrics with those reported in the literature would help the readers to understand the contribution of current work.

(3) The central claim is that the EAP process accelerates polymerization due to an external field. However, from the supporting data it seems the electric field effect was not isolated from standard anionic polymerization. No electrochemical measurements (e.g., chronoamperometry, Faradaic current, or electric-field-dependent kinetics) are provided to demonstrate field-driven acceleration or interface-specific behavior. Plus, to validate the uniqueness of this process, a control using non-electroinitiated ECA coating (e.g., neutral ECA coating without electric field) would be necessary.

(4) Although the authors highlight a high CCD of 6.8 mA/cm² under a reasonable areal capacity (duration of 1 h could be identified based on the figure), the long-term symmetric-cell testing results, either judging from current density or cycle life, are not very competitive compared to those reported in the literature. Although the standard of 4 mA/cm² was mentioned to justify its practical relevance, such demanding condition was not applied in either symmetric cell or pouch cell testing.

(5) Some data is inconsistent and confusing. For example, the cell or interfacial resistance revealed on line 150-152 (Fig. S18 to S19) and on line 205-206 (Fig. S25) differ significantly. It is difficult to judge the soundness of the data based on the current description of the testing conditions (i.e., no information on detailed configuration, solid electrolyte dimensions, fitting results are available).

(6) The high loading testing in full cells is impressive, however, the use of interfacial mending glue (IMG) as a liquid phase additive on the cathode side introduces potential concerns for practical applications.

Reviewer #2

(Remarks to the Author)

The manuscript presents a compelling strategy for interfacial healing in solid-state sodium metal batteries via electroinitiated accelerated polymerization (EAP), achieving a remarkable critical current density of 6.8 mA cm^{-2} and long-term cycling in Ah-level pouch cells without clamping. These results position the work among the top-performing studies in sodium battery research. However, to enhance reader comprehension and facilitate visualization of the dynamic interfacial processes described, the authors are encouraged to provide supplementary videos corresponding to Figures 2b, 3b, 5d, and S16.

Reviewer #3

(Remarks to the Author)

The manuscript titled “Electroinitiated interfacial healing for external pressure-free solid-state sodium metal batteries” introduces an electroinitiated interfacial healing strategy. This approach employs an electroinitiated accelerated polymerization (EAP) process to resolve critical issues in solid-state sodium (Na) metal batteries (SSNMBs), such as interfacial instability and air sensitivity. By leveraging electric field-regulated electrowetting effects, the charged microdroplets preferentially infiltrate cracks and voids on the surfaces of inorganic solid-state electrolytes (e.g., NZZSPO), forming conductive polymer layers that substantially improve interfacial contact and mechanical compatibility. Although the manuscript is well-organized, there are still some questions that need to be clarified (minor revision). Authors are suggested to carefully considering and clarifying the following points for further improving the quality of your manuscript.

1. It is recommended that the authors provide additional experimental parameters related to the electrospray deposition of the IMG monomer, such as the spraying distance, composition of the monomer solution, and whether sodium salt was added. In addition, it would be beneficial to include the electric field or current conditions applied during the electroinitiated accelerated polymerization (EAP) process—for instance, the specific constant current or voltage applied across the symmetric cell and the corresponding duration. Including these details will greatly enhance the reproducibility of the experiments for future research and further support the validity of the data presented in the manuscript.

2. Figure 2b presents the interfacial healing process within a sealed optical cell. It is recommended that the authors include an additional image showing the state of the crack after 10 minutes of exposure to IMG microdroplets, similar to what is shown in Figure S16. This addition would significantly enhance the persuasiveness of the visual evidence. Moreover, it is suggested that the IMG microdroplets be clearly labeled in the figure to help readers easily identify and understand the process being depicted.

3. Figure 2e compares the polymerization time between the conventional drop-coating method and the EAP strategy. It is recommended that Figure 2e be moved to the Supporting Information section, and Figure S17 be placed in the main text instead, as this would allow a more intuitive comparison of the polymerization times between the two methods for the readers. Additionally, the authors are encouraged to include a new figure showing a quantitative analysis curve to emphasize the reported 21.4-fold increase in polymerization rate, which would provide more convincing support than the current presentation in Figure 2e. It is also suggested that the layout of Figure 2 be redesigned to improve visual clarity and overall aesthetic appeal.

4. Figure 3a presents the XRD patterns; it is recommended that the authors add appropriate symbols (e.g., ▲ and ●) in the figure to indicate the main diffraction peaks corresponding to each phase. Additionally, a brief explanation of these diffraction peaks should be included in the main text to help readers better understand the phase composition and structural changes.

5. Figure 3b illustrates the oxidation of Na metal surfaces under blank conditions and the EAP strategy. It appears that there may be a possible misplacement of the third and fourth sodium samples under the EAP condition. The authors are kindly requested to carefully check the image placement to ensure the accuracy of the figure.

6. In the section titled “Uniform sodium deposition behavior,” it is recommended that the authors include, in the Supporting Information, a set of control experimental results without polymer healing treatment. For example, the symmetric cell cycling curves and EIS spectra of the bare Na/NZZSPO interface under the same testing conditions. Although the main text describes the behavior of the control group, presenting the actual data curves would provide a more intuitive and quantitative comparison, clearly highlighting the improvements achieved by the EAP strategy.

7. In Figures 5c and 5e, it is recommended to revise the label “Liquid electrolyte” to “Adding conventional liquid electrolyte” for greater clarity and to accurately reflect the experimental condition described.

8. In Figure 6a, it is recommended to revise the label “Adding conventional electrolytes” to “Adding conventional liquid electrolytes” to help readers clearly associate it with the abbreviation “CLE.” Additionally, the authors are encouraged to include the Coulombic efficiency data for both the blank group and the CLE group to better highlight the superior performance of the EAP strategy. In Figures 6b and 6c, only the data for SSNMBs using the EAP strategy are shown; it is suggested to add corresponding data for the blank and CLE groups for a more comprehensive comparison.

Version 1:

Reviewer comments:

Reviewer #1

(Remarks to the Author)

The authors have properly addressed or at least reasonably discussed the major concerns raised during the initial review and have significantly improved the manuscript. The revised version incorporates new data, expanded discussion, and helpful comparisons that strengthen the work. The reviewer appreciates the authors' comprehensive responses and careful revisions.

At this stage, I have only two minor comments that the authors may wish to clarify to further enhance the manuscript:

(1) The manuscript frequently references finite element modeling (FEM) to simulate droplet spreading behavior and crack healing (e.g., Fig. 1e–g, Supplementary Movies 1 and 2). However, it remains unclear to what extent these simulations are intended to quantitatively predict experimental behavior versus providing qualitative illustration. For instance, while the electrowetting behavior is modeled under an electric field, the connection between the simulation outputs and the measured contact angle changes (from 63.1° to 10.5°) is not explicitly discussed. Clarifying whether experimental parameters (e.g., droplet charge, voltage, surface roughness) were used to constrain the FEM inputs would help reinforce the relevance of these simulations.

(2) In the response letter, the authors cite thermogravimetric analysis (TGA) showing that the residual liquid content in the polymerized IMG layer is less than 0.3 wt%. This is an important result supporting the long-term stability of the cathode interface. However, this finding is only mentioned in the supplementary information and does not appear in the main manuscript. Including a brief description of the TGA result in the main text—along with testing conditions such as temperature range, ramp rate, and carrier gas—would help strengthen this point and address any lingering concerns about potential liquid-phase residues.

These are relatively minor points. I am pleased to recommend acceptance after minor revision.

Reviewer #2

(Remarks to the Author)

The authors have addressed my comments. I can recommend the publication of the manuscript.

Reviewer #3

(Remarks to the Author)

The manuscript meets the publication requirements and is recommended for acceptance

Reviewer #1 (Remarks to the Author):

This manuscript introduces an electroinitiated accelerated polymerization (EAP) strategy that utilizes charged microdroplets to facilitate interfacial healing in solid-state sodium metal batteries (SSNMBs) based on modified NASICON-based solid electrolytes. Particularly, the strategy enables improvements in critical current density (CCD) and long-term cycling stability and unlock pressure-free SSNMBs. The authors present computational analysis along with a wide range of data, including in situ spectroscopies, TOF-SIMS, XPS, FIB-SEM, Nano-CT, and pouch cell testing, in support of their claims. While the work is comprehensive, well-organized and interesting, I have concerns about the mechanistic foundation, novelty compared to existing strategies, and the lack of rigorous benchmarking performances. Unfortunately, these issues significantly limit the manuscript's suitability for publication in Nature Communications. I will elaborate my detailed comments below for the authors' consideration in further improving the manuscript.

(1) The topic is timely and addresses one of the most pressing issues in the commercialization of all-solid-state sodium batteries. However, prior work on interfacial polymer coatings or plastic crystal coatings have demonstrated similar benefits in mitigating voids and cracks in various solid electrolytes (i.e., doi/full/10.1021/acscentsci.6b00321). While the use of ECA and electroinitiated polymerization adds a fresh angle, the novelty is somewhat incremental unless mechanistic advantages are better substantiated.

Response:

We sincerely appreciate the reviewer's thoughtful and forward-looking comments regarding the positioning of our work within the broader landscape of interfacial engineering strategies for solid-state Na metal batteries. While prior studies have indeed demonstrated that soft interface layers such as high-temperature in situ interlayer formation or polymer sandwich architectures can improve physical contact and mechanical compliance (*Zhou W. et al. ACS Cent. Sci. 2017, 3, 52-57; Zhou W. et al. J. Am. Chem. Soc. 2016, 138, 9385-9388.*), we respectfully emphasize that our work

establishes a brand-new approach of electroinitiated accelerated polymerization (EAP) strategy, which introduces several mechanistic and functional advancements that go beyond incremental modification. Our EAP strategy exhibits superior superwetting behavior on the NZZSPO electrolyte compared to the conventional drip-coating method. Under the influence of a high electric field, the IMG microdroplets spread extensively and remained in an ultra-flat configuration on the NZZSPO surface owing to the optimized interfacial tension, which can facilitate the uniform coverage and promote effective subsequent interfacial healing. Our EAP strategy provides direct visualization of this dynamic, self-driven healing behavior, a property beyond the reach of previous drop-cast or thermally applied coatings. In contrast to traditional scraping method or adding extra membranes, our approach leverages electron-transfer-induced polymerization in a low-dew-point environment, which is fully compatible with current commercial battery assembly environment. More importantly, the electrical field-driven initiation at the electrode interface, coupled with electrospray-induced microdroplet charging, ensures rapid, localized, and directional polymerization. Thus, the EAP strategy is inherently more suitable with large-scale commercial deployment compared to conventional alloying-based interfacial modification, interfacial polymer coating, or plastic crystal coating engineering. Our method eliminates the need for high-temperature treatments or reactive metal processing, enabling direct integration into existing solid-state battery fabrication lines with minimal equipment modification.

To substantiate the interface-bonding and mechanical advantages of the ECA-based interfacial layer, we have incorporated new tensile-stretching tests of the polymerized IMG using our EAP strategy in the revised manuscript. The results demonstrate that, upon EAP treatment, the polymerized IMG strongly anchors the ceramic electrolyte to the electrode with an interfacial tensile-shear strength of 4.28 MPa, enabling stable solid-state operation without external clamping, a condition rarely achieved in conventional systems. We further performed atomic force microscopy (AFM) nanoindentation on the healed interfacial layer. The measurements reveal a Young's modulus of 7.29 GPa, which is substantially higher than that of typical SEI layers (0.74 GPa). This unique interfacial structure provides both mechanical robustness and

conformability, effectively suppressing Na dendrite growth by accommodating local stress and redistributing ionic flux. In addition to mechanical healing, our polymerized IMG layer confers unprecedented air stability to both the Na metal and oxide/sulfide-based solid electrolytes (Fig. 3), preserving surface integrity after months of ambient exposure. Moreover, our strategy is implemented without the need for external clamping force (Fig. 6), and is demonstrated in Ah-level pouch cells, a key step toward practical scalability that many earlier interfacial modification strategies do not address.

Page no.5

EAP treatment results in an interfacial tensile-shear strength of 4.28 MPa, providing robust mechanical adhesion that firmly anchors the electrodes to the ceramic electrolyte (**Supplementary Fig. 18**).

The polymerized IMG formed via the EAP strategy exhibits a high Young's modulus of 7.29 GPa (**Supplementary Fig. 37**), significantly surpassing that of the SEI layer after adding CLE (0.74 GPa). Such enhanced mechanical rigidity provides the interfacial healing layer with a superior ability to resist deformation and effectively suppress Na dendrite penetration.

Supplementary Figure 18. Stress-strain curves of EAP strategy and conventional drip-coating method measured by electronic universal testing machines.

Supplementary Figure 37. Indentation curves of (a) the SEI layer on Na metal anode after adding conventional liquid electrolyte (CLE) and (b) the surface of polymerized IMG formed via the EAP strategy enabled by atomic force microscopy analysis.

(2) The absence of stacking pressure is demonstrated as a key breakthrough, nonetheless the interfacial resistance is still quite high after the modification (319 Ohm), although the area is not provided to estimate the ASR. In comparison, it is noticeable that some recent interfacial strategy such as alloying based interface (Na-Pb, Na-Sn, etc.) can achieve near zero resistance while also supporting zero stacking pressure. Considering the key findings of this work, the reviewer would suggest the authors provide a concise overview of key anode issues on current NASICON-based SSNMBs in the introduction part. Also, a comparison in key performance metrics with those reported in the literature would help the readers to understand the contribution of current work.

Response:

We thank the reviewer for the insightful comments and constructive feedback. We agree with Reviewer #1 that stacking-pressure-free operation is a central challenge in the practical deployment of ceramic electrolyte-based solid-state Li/Na metal batteries. Unlike introducing alloying-based (e.g., Pb, Sn, Al, and Zn) interfaces, those systems typically involve reactive wetting, require careful stoichiometry control and complex preparation conditions, and in many cases lack polymeric sealing or environmental

stability, especially in large-format cell configurations. Introducing Na-Pb or Na-Sn alloy interfaces may precipitate the electrochemically active species at the anode after long-term cycling, while the interface modified by our EAP strategy is electrochemically inert, structurally stable, and supports long-term plating/stripping processes without dendrite formation. Furthermore, our modified interface provides mechanical adhesion and air-stability, enabling Ah-level pouch cell operation under ambient conditions without stacking pressure. For our EAP strategy, the measured interfacial resistance of $\sim 318.9 \Omega$ reported in Na metal-based symmetric cells without using Au-based ion-blocking electrode corresponds to a standard pellet size of 16.0 mm in diameter, which yields an area-specific resistance (ASR) of $158.7 \Omega \cdot \text{cm}^{-2}$ at room temperature without any applied stacking pressure or thermal activation, much lower than that of conventional drip-coating method ($218.8 \Omega \cdot \text{cm}^{-2}$) or adding conventional liquid electrolyte ($2590.9 \Omega \cdot \text{cm}^{-2}$).

According to the reviewer's suggestion, we have now added a concise summary of the key challenges of anode associated with solid-state ceramic electrolytes in the Introduction section. To help readers better appreciate the impact of our work, we have added a comparative summary table and figures in the revised manuscript (Supplementary Figs. 58, 59, and Table 3), benchmarking our EAP strategy-enabled performance metrics, such as CCD (6.8 mA cm^{-2}), CE ($>99.8\%$), rate performance (15 C), Ah-level pouch cell, and pressure-free operation, against representative state-of-the-art strategies, including alloying layers, melt-infiltrated interfaces, polymer-modified surfaces, and so on.

Page no.2

However, the transition from liquid-solid interfaces to solid-solid interfaces poses significant challenges, where issues related to unfavorable mechanical problems, poor metal anode wettability, interfacial instability, and physical contact barriers become critical constraints⁷⁻⁹. Despite notable advancements, the risk of anode-side dendrite propagation remains non-negligible, posing a persistent threat to interfacial stability, internal short circuiting, and catastrophic battery failure.

The accumulation of passivation products on the surface of OSEs further diminishes wettability and suppresses ionic conductivity at the metal anode interface, thereby enlarging the space charge layer, hindering ion transport, and increasing the interfacial resistance²².

Similarly, the EAP strategy is also applicable to other solid-state conductors, among which NaPSCI-based sulfide solid electrolytes show significantly improved performance in symmetric cells and SSNMBs with NFM cathodes (**Supplementary Figs. 56 and 57**), outperforming reported literature (**Supplementary Figs. 58, 59, and Table 3**).

Supplementary Table 3. Comparison of the performance for various reported interfacial optimization strategies.^{S21, S22, S38-S42}

Optimization strategy	External pressure	CCD value	Coulombic efficiency	Rate performance	Pouch cell
Mechano-electrochemical healing	35 MPa	/	99.40%	5.0 C	/
Ion-conductive polymer layer	/	1.58 mA cm ⁻²	98.19%	5.0 C	/
Polymer interface coating	/	1.1 mA cm ⁻²	/	1.0 C	/
Liquid metal gallium	50 MPa	1.7 mA cm ⁻²	~100%	~1.5 C	/
Na-Au alloy interface layer	0 MPa	0.8 mA cm ⁻²	95.20%	5.0 C	Single-layer cathode
Polymer interlayer	0 MPa	1.4 mA cm ⁻²	91.80%	2.0 C	/

KF electron-blocking interlayer	0 MPa	1.0	~99.99%	2.0 C	/
		mA cm^{-2}			
Lithiophilic layer	0 MPa	2.4	99.50%	3.0 C	/
		mA cm^{-2}			
Our EAP strategy	0 MPa	6.8	99.80%	15.0 C	1.0 Ah, 100 cycles
		mA cm^{-2}			

Supplementary Figure 58. Comparison of critical current density value of various inorganic solid-state electrolytes using the high-pressure mold and without external stack pressure.^{S1-S15}

Supplementary Figure 59. Comparison of capacity retention and Coulombic efficiency of the reported works within the last 5 years.^{S16-S37}

S38. Yang, G. *et al.* A Bridge between Ceramics Electrolyte and Interface Layer to Fast Li^+ Transfer for Low Interface Impedance Solid-State Batteries. *Adv. Funct. Mater.* **33**, 2211387 (2023).

- S39. Miao, X. *et al.* Isotropic Sulfurized Polyacrylonitrile Interlayer with Homogeneous Na⁺ Flux Dynamics for Solid-State Na Metal Batteries. *Adv. Energy Mater.* **11**, 2003469 (2021).
- S40. Lee, S. *et al.* Mechano-Electrochemical Healing at the Interphase Between LiNi_{0.8}Co_{0.1}Mn_{0.1}O₂ and Li₆PS₅Cl in All-Solid-State Batteries. *Adv. Energy Mater.* 2405782 (2025).
- S41. Li, D. *et al.* Atomically bonding Na anodes with metallized ceramic electrolytes by ultrasound welding for high-energy/power solid-state sodium metal batteries. *Carbon Energy* **5**, e299, (2023).
- S42. Meng, L., Zhang, Y., Zhou, X., Lei, M. & Li, C. Li₂CO₃-affiliative mechanism for air-accessible interface engineering of garnet electrolyte via facile liquid metal painting. *Nat. Commun.* **11**, 3716 (2020).

(3) The central claim is that the EAP process accelerates polymerization due to an external field. However, from the supporting data it seems the electric field effect was not isolated from standard anionic polymerization. No electrochemical measurements (e.g., chronoamperometry, Faradaic current, or electric-field-dependent kinetics) are provided to demonstrate field-driven acceleration or interface-specific behavior. Plus, to validate the uniqueness of this process, a control using non-electroinitiated ECA coating (e.g., neutral ECA coating without electric field) would be necessary.

Response:

Thank the reviewer for useful advice. Conventionally, the anionic polymerization involves the use of a strong nucleophile (such as alkali metals, organic compounds, bases, Grignard reagents, and electron donors) to initiate the polymerization of monomers with electron-withdrawing groups. A strong nucleophile (the initiator) attacks the electron-deficient carbon-carbon double bond of a monomer, forming a carbanion. The carbanion reacts with another monomer molecule, transferring the negative charge and forming a new carbanion. The process repeats in a chain reaction,

leading to the growth of the polymer chain (*Angew. Chem. Int. Ed.* 2021, 60, 16487; *Chem. Eng. J.* 2021, 404, 126470.). For the conventional drip-coating method, the polymerization of ECA is triggered by the moisture under ambient conditions. For battery assembly, especially for alkali metal batteries, a strict water-free environment is required, such as a controlled dry room with a dew point below $-35\text{ }^{\circ}\text{C}$ (as described in the Methods section). Under such low-moisture conditions, spontaneous polymerization via conventional mechanisms is severely suppressed, as confirmed by our control experiments showing that conventional drip-coating method remains unpolymerized for hours in the absence of an applied field (Figure 2c). In sharp contrast, our EAP strategy relies on electroinitiated charge transfer at the electrode-monomer interface. When in contact with the electrode, the electron transfer process becomes thermodynamically favorable in charged IMG microdroplets obtained from the electrospray process, where electrons are transferred to the electrophilic ECA monomers and generate carbanions, which further react with the successive monomers to polymerize via a chain growth mechanism and increase the polymerization rate by 21.4 times (Figure 2d and 2e).

In response to the reviewer's suggestion, we have included a series of comparative experiments in the revised manuscript using conventional drip-coating method without electric field. Compared with adding conventional liquid electrolytes or conventional drip-coating method, our EAP strategies exhibit the lowest area-specific resistance (Supplementary Fig. 29). From Supplementary Figs. 21 and 22, chronoamperometry combined with two AC impedance measurements were used to investigate the Na^+ transference number, where the symmetrical cell modified by conventional drip-coating method delivers a Na^+ transference number of 0.81, lower than that of our EAP strategy (0.90). The linear sweep voltammetry was applied to determine the electrochemical windows of various strategy, where both conventional drip-coating method and EAP strategy exhibit a similar high potential window up to 5.0 V due to the excellent antioxidant properties of NZZSPO electrolytes (Supplementary Fig. 25). When paired with NFM cathode, adding conventional liquid electrolytes proves ineffective at elevated active material loading, and conventional drip-coating method often results in

unstable discharging capacities with pronounced fluctuations under high-loading conditions. These results provide strong evidence that our EAP strategy is indispensable for both initiating rapid polymerization and enabling interfacial wetting and healing under dry room conditions, resulting in a high Na^+ transference number and improved electrochemical performance.

Page no.5

Benefiting from the high ionic conductivity after matching the NZZSPO conductor and enhanced physical contact by EAP strategy, the Na^+ transference number was calculated to be 0.90 approaching unity (**Supplementary Fig. 21**), which is much higher than that of conventional drip-coating method (**Supplementary Fig. 22**), indicating more efficient Na^+ transport across the interface.

Supplementary Figure 22. Chronoamperometry profile of symmetric cell optimized by conventional drip-coating method and corresponding EIS spectra (inset).

Supplementary Figure 25. LSV curves of bare conventional liquid electrolyte and oxide solid electrolytes optimized by conventional drip-coating method and EAP strategy.

As confirmed by EIS measurements, the overall resistance ($158.7 \Omega \text{ cm}^{-2}$) of the cell optimized by IMG is much lower than that of CLE ($2590.9 \Omega \text{ cm}^{-2}$) and conventional drip-coating method ($218.8 \Omega \text{ cm}^{-2}$, **Supplementary Fig. 28**), further demonstrating the successful elimination of interfacial issues between Na metal and NZZSPO electrolyte.

Supplementary Figure 28. Nyquist plots of the symmetric Na metal solid-state cell optimized by EAP strategy, conventional drip-coating method, and conventional liquid electrolyte. The inset shows a zoomed-in view of the high-frequency region.

Adding CLE proves ineffective at elevated active material loading, and conventional drip-coating method often results in unstable discharging capacities with pronounced fluctuations under high-loading conditions (**Supplementary Figs. 48 and 49**).

Supplementary Figure 48. Cycling performance of SSNMBs paired with NVP electrode optimized by (a) adding CLE and (b) conventional drip-coating method with different active material loadings.

Supplementary Figure 49. Corresponding Coulombic efficiency of SSNMBs optimized by (a) adding CLE and (b) conventional drip-coating method with different active material loadings.

(4) Although the authors highlight a high CCD of 6.8 mA/cm² under a reasonable areal capacity (duration of 1 h could be identified based on the figure), the long-term symmetric-cell testing results, either judging from current density or cycle life, are not very competitive compared to those reported in the literature. Although the standard of 4 mA/cm² was mentioned to justify its practical relevance, such demanding condition was not applied in either symmetric cell or pouch cell testing.

Response:

Thank the review for the insightful comments. While we originally emphasized the achievement of a critical current density (CCD) of 6.8 mA cm⁻² under a realistic areal capacity, we fully recognize the reviewer's concern that long-term symmetric cell and full cell testing at practically relevant current densities (e.g., ≥ 4 mA cm⁻²) was not adequately demonstrated in the initial version. In our revised manuscript, we have added new Na metal-based symmetric-cell cycling data under an increasing current densities of 1.0, 2.0, 4.0, 5.0, and 6.0 mA cm⁻², which exhibits a stable performance with low overpotential, without any dendrite-induced micro-short circuit and voltage fluctuation signals. Furthermore, the full cells paired with NFM cathode (13.77 mg cm⁻²) exhibits excellent cycling performance over 150 cycles under a current density of 5.25 mg cm⁻² (3.0 C). These results confirm that our EAP strategy maintains robust interfacial integrity and supports fast Na⁺ transport under stringent operating conditions. We have updated the relevant sections in the main text and figure legends accordingly.

Page no.6

Furthermore, the symmetric cell demonstrates stable rate performance from 1.0 to 6.0 mA cm⁻² characterized by a moderate increase in overpotential without noticeable fluctuations (**Supplementary Fig. 30**).

A stable cycling performance is achieved at a high current density of 5.25 mA cm⁻² (**Supplementary Fig. 54**).

Supplementary Figure 30. Rate capability under various current densities from 1.0 to 6.0 mA cm⁻².

Supplementary Figure 54. Cycling performance of SSNMBs paired with NFM cathode optimized by EAP strategy under a current density of 5.25 mA cm⁻².

(5) Some data is inconsistent and confusing. For example, the cell or interfacial resistance revealed on line 150-152 (Fig. S18 to S19) and on line 205-206 (Fig. S25) differ significantly. It is difficult to judge the soundness of the data based on the current description of the testing conditions (i.e., no information on detailed configuration, solid electrolyte dimensions, fitting results are available).

Response:

Thanks for your careful review and valuable feedback. To clarify, the data in Supplementary Figs. 19 and 20 were obtained using the Au/NZZSPO/Au blocking electrode symmetric cells to measure the temperature-dependent bulk ionic conductivity of the NZZSPO solid electrolyte with different optimization strategies. The impedance of solid-state electrolytes with ion-blocking electrode (Au) was obtained to only investigate the responses of ionic transport between solid electrolytes and modified interface of electrolytes due to the blocking of ionic transport at the side of electrode interface. This method, commonly adopted in prior literature on solid-state electrolytes (*Han X. et al. Nat. Mater. 2017, 16, 572-579; Wang S. et al. J. Am. Chem. Soc. 2018, 140, 250-257; Deng. T. et al. Nat. Nanotechnol. 2022, 17, 269-277; Jung S.-K. et al. Nat Commun. 2022, 13, 7638; Kong X. et al. Nat Commun 2024, 15, 7247.*), isolates the intrinsic transport properties of the solid electrolyte and eliminates contributions from electrode interface effects. This test method is mainly used to show the changes in ionic conductivity of the solid electrolyte itself under different optimization methods. In contrast, Supplementary Fig. 29 presents the EIS profiles collected from Na metal-based symmetric cells, where the total resistance includes not only the bulk resistance of the solid electrolyte, but also the interfacial resistance between Na metal and NZZSPO, as well as possible contributions from interphase layers and contact resistance. As a result, these two measurements inherently reflect different physical phenomena and optimization results, which are not suitable for direct comparison. We also modified the corresponding figure legends and Methods section to avoid ambiguity for readers.

Page no.11

A thin layer of Au (thickness ~300 nm) was sputtered on one side of stainless steel pellet using a sputter coater (Saintins, JS-1600) and acted as a blocking electrode. AC impedance analysis from 7 MHz to 1 Hz with an amplitude of 10 mV was used to measure the ionic conductivity (BioLogic VMP3 potentiostat) with the Au/SSE/Au symmetric cells based on the following Equation:

$$\sigma_t = \frac{L}{R_t S}$$

where S , R_t , L , and σ_t represent the surface area of Au electrode, resistance value, thickness of NZZSPO pellet, and ionic conductivity.

Supplementary Figure 19. a,b, Nyquist plots of the solid-state cells with ceramic electrolytes and Au-blocking electrodes modified by adding (a) conventional liquid electrolytes and (b) IMG using EAP strategy measured at different temperatures.

(6) The high loading testing in full cells is impressive, however, the use of interfacial mending glue (IMG) as a liquid phase additive on the cathode side introduces potential concerns for practical applications.

Response:

We sincerely appreciate the reviewer's recognition of the high-loading performance demonstrated in our full-cell configurations, as well as the insightful concern regarding the use of EAP strategy at the cathode side and its implications for practical applications. In our strategy, although IMG is initially introduced in the form of charged microdroplets via electrospray, it undergoes rapid in-situ electroinitiated polymerization upon contact with the electrode under an applied electric field. This process results in the formation of a solidified, ionically conductive, and mechanically robust interfacial layer, which effectively reduces and minimizes the residual liquid phase in the battery. Thermogravimetric analysis (TGA) reveals that the residual liquid content in the EAP-treated interface is minimal, accounting for less than 0.3% by weight, indicating the formation of a predominantly solid interfacial healing layer. This is in sharp contrast to conventional liquid electrolyte addition, which often leave around 6.9% of undesired liquid residues that may compromise long-term stability. Moreover, the IMG is applied in a highly localized and minimal manner, confined strictly to the electrolyte-electrode interface, connecting the cathode and the electrolyte. From SEM images of the cycled NFM cathode, our EAP strategy reveals a smooth and crack-free

surface, whereas the cathode treated with conventional liquid electrolytes exhibits clear surface roughening and crack formation. This comparison underscores that our interface engineering approach not only enhances interfacial contact and mechanical integrity but also preserves the structural stability of the cathode throughout long-term cycling processes.

Page no.9

Supplementary Fig. 53 exhibited intuitionistic particle cracks in the cycled NFM cathode for SSNMBs optimized by adding CLE, whereas EAP strategy still maintained its integrity morphology without cracks even after 50 cycles.

Supplementary Figure 53. SEM images of NFM cathode for SSNMBs (a) optimized by adding CLE and (b) using EAP strategy after 50 cycles.

Supplementary Figure 12. Thermogravimetric analysis curves of NZZSPO with interfacial optimization strategy including adding conventional liquid electrolyte, conventional drip-coating method, and electroinitiated accelerated polymerization

strategy.

Reviewer #2 (Remarks to the Author):

The manuscript presents a compelling strategy for interfacial healing in solid-state sodium metal batteries via electroinitiated accelerated polymerization (EAP), achieving a remarkable critical current density of 6.8 mA cm^{-2} and long-term cycling in Ah-level pouch cells without clamping. These results position the work among the top-performing studies in sodium battery research. However, to enhance reader comprehension and facilitate visualization of the dynamic interfacial processes described, the authors are encouraged to provide supplementary videos corresponding to Figures 2b, 3b, 5d, and S16.

Response:

We thank Reviewer #2 for the positive and encouraging comments regarding our manuscript. We are pleased that the reviewer recognized the significance of our electroinitiated accelerated polymerization (EAP) strategy in enabling exceptional interfacial stability, achieving a high critical current density and demonstrating long-term cycling performance in Ah-level pouch cells without clamping.

In response to the reviewer's insightful suggestion to enhance reader comprehension and facilitate visualization of the dynamic interfacial healing process, we have now provided supplementary video files corresponding to Figs. 1e, 1g, 2b, 5d, and Supplementary Fig. 13. These videos offer direct visual evidence of the real-time interfacial evolution, including the finite element simulation results, charged microdroplet spreading, crack-filling behavior, and suppression of dendrite propagation during cycling. Figure 3b shows that our strategy can improve the stability of Na metal in air, and we were only able to capture time-resolved photographic images at specific intervals. These sequential images included in our previously submitted and revised manuscript effectively demonstrate the progressive oxidation behavior and clearly highlight the enhanced air stability imparted by our EAP strategy. We believe these supplementary materials will significantly aid readers in grasping the mechanisms

and effectiveness of our EAP strategy. The corresponding Description of Supplementary Videos and Supplementary Information section have been updated accordingly.

Additional Supplementary Files:

Supplementary Movie 1. Simulated coating process by finite element simulation through conventional drip-coating method (left) and our electroinitiated accelerated polymerization strategy (right).

Supplementary Movie 2. Simulated flow process of our electroinitiated accelerated polymerization strategy by finite element simulation.

Supplementary Movie 3. Crack healing process using our electroinitiated accelerated polymerization strategy taken with microscopy.

Supplementary Movie 4. *In-situ* optical microscopy observations of oxide solid electrolytes modified by adding conventional liquid electrolyte.

Supplementary Movie 5. *In-situ* optical microscopy observations of oxide solid electrolytes modified by our electroinitiated accelerated polymerization strategy.

Reviewer #3 (Remarks to the Author):

The manuscript titled “Electroinitiated interfacial healing for external pressure-free solid-state sodium metal batteries” introduces an electroinitiated interfacial healing strategy. This approach employs an electroinitiated accelerated polymerization (EAP) process to resolve critical issues in solid-state sodium (Na) metal batteries (SSNMBs), such as interfacial instability and air sensitivity. By leveraging electric field-regulated electrowetting effects, the charged microdroplets preferentially infiltrate cracks and voids on the surfaces of inorganic solid-state electrolytes (e.g., NZZSPO), forming conductive polymer layers that substantially improve interfacial contact and mechanical compatibility. Although the manuscript is well-organized, there are still some questions that need to be clarified (minor revision). Authors are suggested to carefully considering and clarifying the following points for further improving the quality of your manuscript.

1. It is recommended that the authors provide additional experimental parameters related to the electrospray deposition of the IMG monomer, such as the spraying distance, composition of the monomer solution, and whether sodium salt was added. In addition, it would be beneficial to include the electric field or current conditions applied during the electroinitiated accelerated polymerization (EAP) process—for instance, the specific constant current or voltage applied across the symmetric cell and the corresponding duration. Including these details will greatly enhance the reproducibility of the experiments for future research and further support the validity of the data presented in the manuscript.

Response:

We greatly appreciate Reviewer #3's thoughtful suggestion regarding the inclusion of detailed experimental parameters to improve the reproducibility and transparency of our electroinitiated accelerated polymerization strategy. We have comprehensively revised the Methods section to incorporate the following key parameters related to our EAP strategy: A high-voltage electric field of +15.0 kV was applied to the spraying needle, while the NZZSPO pellet substrate was connected to -2.0 kV and mounted on a vacuum chuck to ensure effective droplet deposition and uniform coverage. The spraying distance was maintained at approximately 8-15 cm, allowing for optimal electrowetting and spreading of IMG microdroplets at the electrolyte interface. The IMG flow rate was precisely controlled at 0.06 mm min^{-1} , where a single side of coin cells takes 1-3 seconds and a single side of pouch cells takes 5-7 seconds. All corresponding information has been clearly integrated into the Methods section of revised manuscript.

Page no.10

The improved liquid electrostatic spraying gun (LVOBO, BO-818) was used for EAP processes. A high-voltage electric field of +15.0 kV was applied to the spraying needle, while the NZZSPO pellet substrate was connected to -2.0 kV and mounted on a vacuum chuck to ensure effective droplet deposition and uniform coverage. The spraying

distance was maintained at approximately 8-15 cm, allowing for optimal electrowetting and spreading of IMG microdroplets at the electrolyte interface. The IMG flow rate was precisely controlled at 0.06 mm min^{-1} , where a single side of coin cells takes 1-3 seconds and a single side of pouch cells takes 5-7 seconds. The entire process occurred during battery assembly, with IMG covering the entire NZZSPO electrolyte surface and then quickly sandwiched between the cathode electrode and Na metal anode.

2. Figure 2b presents the interfacial healing process within a sealed optical cell. It is recommended that the authors include an additional image showing the state of the crack after 10 minutes of exposure to IMG microdroplets, similar to what is shown in Figure S16. This addition would significantly enhance the persuasiveness of the visual evidence. Moreover, it is suggested that the IMG microdroplets be clearly labeled in the figure to help readers easily identify and understand the process being depicted.

Response:

We are grateful for reviewer's insightful comments and valuable feedback. In the original manuscript, Fig. 2b and Supplementary Fig. 16 indeed depicted the same interfacial crack healing process on the surface of NZZSPO enabled by the EAP strategy. The time-lapse image capturing the crack state at 10 minutes was placed in the Supporting Information due to initial typesetting constraints. In the revised manuscript, we have now relocated the complete sequence of the crack healing process to Fig. 2b and Supplementary Movie 3, thereby presenting a more continuous and persuasive visual narrative of the interfacial healing mechanism. Moreover, we have revised Fig. 2b to clearly label the IMG microdroplets, allowing readers to readily identify their position and role in the electrowetting-driven infiltration and polymerization process.

Fig. 2b. Photographs of crack healing process using EAP strategy taken with microscopy.

3. Figure 2e compares the polymerization time between the conventional drop-coating method and the EAP strategy. It is recommended that Figure 2e be moved to the Supporting Information section, and Figure S17 be placed in the main text instead, as this would allow a more intuitive comparison of the polymerization times between the two methods for the readers. Additionally, the authors are encouraged to include a new figure showing a quantitative analysis curve to emphasize the reported 21.4-fold increase in polymerization rate, which would provide more convincing support than the current presentation in Figure 2e. It is also suggested that the layout of Figure 2 be redesigned to improve visual clarity and overall aesthetic appeal.

Response:

We appreciate reviewer's valuable feedback. In our revised manuscript, we have redesigned the layout of Fig. 2 to enhance its overall visual clarity, narrative coherence, and aesthetic appeal. Specifically, we have relocated the original Fig. 2e to the Supporting Information and correspondingly promoted Supplementary Fig. 17 to the main text, which provides a more intuitive and direct comparison of the polymerization kinetics between the conventional drop-coating method and our EAP strategy. To further strengthen the quantitative rigor of our claim regarding the 21.4-fold increase in polymerization rate, we have added a new figure panel illustrating comparative curves of the degree of polymerization derived from the time-dependent FT-IR spectral evolution of the disappearance of C=C and =CH₂ bonds (Fig. 2e).

Fig. 2. In-situ FT-IR spectroscopy using (c) conventional drip-coating method and (d) EAP strategy. e, Corresponding changes in the degree of polymerization over time for various strategies.

4. Figure 3a presents the XRD patterns; it is recommended that the authors add appropriate symbols (e.g., ▲ and ●) in the figure to indicate the main diffraction peaks corresponding to each phase. Additionally, a brief explanation of these diffraction peaks should be included in the main text to help readers better understand the phase composition and structural changes.

Response:

We appreciate reviewer's valuable comments and suggestions. We have annotated the XRD patterns in Fig. 3a with distinct symbols to clearly indicate the main diffraction peaks corresponding to key phases of Na₂O₂, Na₂CO₃, and NaOH. These markers facilitate rapid identification of phase transformations resulting from air exposure. We have incorporated an informative explanation in the main text to describe the origin and significance of these diffraction peaks. Specifically, we clarify that the emergence of peaks corresponding to Na₂O₂, Na₂CO₃, and NaOH signifies the surface degradation of bare NZZSPO after prolonged exposure to ambient air, while the absence of these peaks in the EAP-treated samples confirms the effectiveness of our strategy in maintaining phase integrity and achieving air stability.

As shown in **Fig. 3a**, the X-ray diffraction (XRD) curves confirmed that the sodiophobic sodium carbonate, sodium peroxide, and sodium hydroxides (NaOH) species were formed on the surface of NZZSPO due to the severe air corrosion caused by prolonged exposure to ambient air for more than half a year, which was not conducive to Na^+ transfer in OSEs.

Fig. 3a. XRD curves of various NZZSPO after exposure to ambient air for half a year.

5. Figure 3b illustrates the oxidation of Na metal surfaces under blank conditions and the EAP strategy. It appears that there may be a possible misplacement of the third and fourth sodium samples under the EAP condition. The authors are kindly requested to carefully check the image placement to ensure the accuracy of the figure.

Response:

We appreciate reviewer's feedback, which has helped us further improve the rigor and accuracy of our data presentation. In the revised manuscript, Fig. 3b has been corrected to accurately reflect the progression of surface oxidation across all Na metal anode. The corrected sequence now clearly demonstrates the superior air stability of Na metal surfaces treated via the EAP strategy compared to the untreated counterparts, as evidenced by the sustained metallic luster and the absence of surface passivation.

Fig. 3b. Photographs of Na disc with and without polymerized IMG exposed to air at various times.

6. In the section titled “Uniform sodium deposition behavior,” it is recommended that the authors include, in the Supporting Information, a set of control experimental results without polymer healing treatment. For example, the symmetric cell cycling curves and EIS spectra of the bare Na/NZZSPO interface under the same testing conditions. Although the main text describes the behavior of the control group, presenting the actual data curves would provide a more intuitive and quantitative comparison, clearly highlighting the improvements achieved by the EAP strategy.

Response:

We greatly appreciate the reviewer’s insightful recommendation to provide direct comparative data for the control group without polymer healing treatment, which offers a more comprehensive and quantitative perspective on the performance enhancement delivered by the EAP strategy. For the symmetric cell with bare NZZSPO, a dendrite-induced short circuit occurs after only a few cycles (Figure 4a), primarily due to the uneven Na deposition arising from poor interfacial contact. EIS spectra of the symmetric cell with bare NZZSPO is provided in Supplementary Figure 29. Among all

evaluated configurations, the NZZSPO optimized by conventional liquid electrolytes (5206.7 Ω) exhibits the highest resistance value compared to that of conventional drip-coating method (439.7 Ω) and our EAP strategy (318.9 Ω), underscoring the critical importance of effective interface engineering for achieving stable electrochemical performance. All corresponding figure references and descriptions have been updated accordingly. We believe this supplemental data will further reinforce the robustness and effectiveness of the proposed interfacial engineering strategy.

Page no.6

As confirmed by EIS measurements, the overall resistance (158.7 $\Omega \text{ cm}^{-2}$) of the cell optimized by IMG is much lower than that of CLE (2590.9 $\Omega \text{ cm}^{-2}$) and conventional drip-coating method (218.8 $\Omega \text{ cm}^{-2}$, **Supplementary Fig. 28**), further demonstrating the successful elimination of interfacial issues between Na metal and NZZSPO electrolyte.

Figure 4a. Voltage-time profiles of the Na metal-based solid-state symmetric cells at the current density of 0.5 mA cm⁻².

Supplementary Figure 28. Nyquist plots of the symmetric Na metal solid-state cell optimized by EAP strategy, conventional drip-coating method, and conventional liquid electrolyte. The inset shows a zoomed-in view of the high-frequency region.

7. In Figures 5c and 5e, it is recommended to revise the label “Liquid electrolyte” to “Adding conventional liquid electrolyte” for greater clarity and to accurately reflect the experimental condition described.

Response:

We greatly appreciate the reviewer’s valuable comments. In accordance with the recommendation, we have revised the label “Liquid electrolyte” to “Adding conventional liquid electrolyte” in Fig. 5c and 5e. This updated terminology more accurately reflects the experimental conditions described in the revised manuscript, wherein a conventional carbonate-based liquid electrolyte was introduced at the interface to serve as a comparative strategy.

Figure 5c. SEM images of cycled oxide solid electrolytes, where interfaces are optimized by the CLE (top) and EAP strategy (bottom) at a current density of 3.0 mA cm^{-2} . Scale bars, $10 \text{ }\mu\text{m}$.

Figure 5e. FIB-SEM images of oxide solid electrolytes after cycling using different interfacial optimization methods.

8. In Figure 6a, it is recommended to revise the label “Adding conventional electrolytes” to “Adding conventional liquid electrolytes” to help readers clearly associate it with the abbreviation “CLE.” Additionally, the authors are encouraged to include the Coulombic efficiency data for both the blank group and the CLE group to better highlight the superior performance of the EAP strategy. In Figures 6b and 6c, only the data for SSNMBs using the EAP strategy are shown; it is suggested to add corresponding data for the blank and CLE groups for a more comprehensive comparison.

Response:

Thank the reviewer for the valuable comments. As suggested, we have revised the label “Adding conventional electrolytes” to “Adding conventional liquid electrolytes” in Fig. 6a. We have now included the comparison of Coulombic efficiency for the cells modified by adding conventional liquid electrolytes or our EAP strategy in the Supplementary Information. Furthermore, we have expanded Fig. 6b and 6c by incorporating the corresponding capacity data for the cells modified by adding conventional liquid electrolytes and conventional drip-coating method under identical testing conditions. Adding conventional liquid electrolytes proves ineffective at elevated active material loading, and conventional drip-coating method often results in unstable discharging capacities with pronounced fluctuations under high-loading conditions. These additions enable a more comprehensive and quantitative evaluation of the electrochemical performance, further highlighting the superiority of the EAP-engineered interface in terms of ionic transport, interfacial contact, and long-term reversibility.

Page no.9

Adding CLE proves ineffective at elevated active material loading, and conventional drip-coating method often results in unstable discharging capacities with pronounced fluctuations under high-loading conditions (Supplementary Fig. 48 and 49).

Figure 6a. Long-term cycling stability of SSNMBs paired with NVP cathode at 1.0 C.

Supplementary Figure 48. Cycling performance of SSNMBs paired with NVP electrode optimized by (a) adding CLE and (b) conventional drip-coating method with different active material loadings.

Supplementary Figure 49. Corresponding Coulombic efficiency of SSNMBs optimized by (a) adding CLE and (b) conventional drip-coating method with different active material loadings.

Reviewer #1 (Remarks to the Author):

The authors have properly addressed or at least reasonably discussed the major concerns raised during the initial review and have significantly improved the manuscript. The revised version incorporates new data, expanded discussion, and helpful comparisons that strengthen the work. The reviewer appreciates the authors' comprehensive responses and careful revisions.

At this stage, I have only two minor comments that the authors may wish to clarify to further enhance the manuscript:

(1) The manuscript frequently references finite element modeling (FEM) to simulate droplet spreading behavior and crack healing (e.g., Fig. 1e–g, Supplementary Movies 1 and 2). However, it remains unclear to what extent these simulations are intended to quantitatively predict experimental behavior versus providing qualitative illustration. For instance, while the electrowetting behavior is modeled under an electric field, the connection between the simulation outputs and the measured contact angle changes (from 63.1° to 10.5°) is not explicitly discussed. Clarifying whether experimental parameters (e.g., droplet charge, voltage, surface roughness) were used to constrain the FEM inputs would help reinforce the relevance of these simulations.

Response:

We thank the reviewer for this valuable comment. In the revised manuscript, we have clarified in the “Finite Element Simulation” subsection of the Methods that the FEM analysis was designed to provide a qualitatively informed yet illustrative representation of droplet spreading and crack healing with EAP strategy. To enhance the connection between simulation and experimental observations, We have improved the Method section and added important parameters used in the FEM simulation, including the size of the model, droplet density, applied voltage, surface roughness, and initial contact angle. These additions further confirm the relevance of the FEM results while maintaining their usefulness as a visual aid to elucidate the underlying mechanisms.

Page no.13

The corresponding finite element simulations and analysis were designed to provide a qualitative and illustrative representation of droplet spreading and crack healing processes under electrowetting conditions.

The droplet density is 1.41 kg m^{-3} with the viscosity of 9.96 mPa s . The applied electric field is achieved by adjusting the contact angle and surface tension of the droplet with the initial contact angle of 10.5° and 63.1° for EAP strategy and conventional drip-coating method. The surface roughness of the receiving bottom is set to 48.7 nm based on the root mean square roughness of OSEs. Furthermore, a two-dimensional finite element model has been established to investigate the crack healing process at the electrode-electrolyte interface under the electric field with the Nernst-Planck and the Laminar flow interface. The simulation area has a height of $7.0 \text{ }\mu\text{m}$ and a width of $8.0 \text{ }\mu\text{m}$. The mesh is chosen to be triangular or tetrahedron-based while

using an increasing refinement toward the electrode bands. The adsorption of IMG on the surface of OSE is determined by the caused by concentration changes, migration in the electric field, and convection caused by density changes, as shown in following equations:

$$\frac{\partial c_i}{\partial t} = -\nabla \cdot J_i \quad (13)$$

$$J_i = -D_i \nabla c_i - z_i \mu_i F c_i E + c_i \vec{v} \quad (14)$$

$$\mu_i = \frac{D_i}{RT} \quad (15)$$

$$\rho \frac{\partial \vec{v}}{\partial t} + \rho(\vec{v} \cdot \nabla) \vec{v} = -\nabla p + \mu \nabla^2 \vec{v} + \rho g \quad (16)$$

$$\rho \nabla \cdot \vec{v} = 0 \quad (17)$$

where C_i is the concentration, J_i is the flux vector, Z_i is the charge number, μ_i is the ion mobility, F is the Faraday constant, D_i is the diffusion coefficient, \vec{v} is the velocity field, p is the pressure of the electrolyte, ρ is the density, and μ is the dynamic viscosity. In the simulation, the gravitational acceleration is set to 9.8 m s^{-2} , and the Na^+ diffusion coefficient is taken as $1.33 \times 10^{-9} \text{ m}^2 \text{ s}^{-1}$. The Faraday constant is 96485 C mol^{-1} with the charge number of 1. The dynamic viscosity of the droplet is 9.96 mPa s , while the potentials applied at the top and bottom boundaries were 15.0 kV and -2.0 kV , respectively.

(2) In the response letter, the authors cite thermogravimetric analysis (TGA) showing that the residual liquid content in the polymerized IMG layer is less than 0.3 wt%. This is an important result supporting the long-term stability of the cathode interface. However, this finding is only mentioned in the supplementary information and does not appear in the main manuscript. Including a brief description of the TGA result in the main text—along with testing conditions such as temperature range, ramp rate, and carrier gas—would help strengthen this point and address any lingering concerns about potential liquid-phase residues.

Response:

Thank the reviewer for useful advice. In the revised manuscript, we have incorporated a concise yet explicit description of the TGA analysis in the section of “Electroinitiated accelerated interfacial healing”, highlighting that the residual liquid content after EAP polymerization is less than 0.3 wt%, far below that of conventional drip-coating method. This indicates the formation of a predominantly solid interfacial healing layer that firmly bridges the electrodes and solid-state electrolyte. To ensure reproducibility and methodological transparency, we have also detailed the full testing parameters, such as temperature range, heating rate, and carrier gas, in the “Material Characterization” subsection of the Methods. We believe that including these details directly in the main text addresses the reviewer’s concern and further reinforces the practicality and reliability of our EAP strategy for solid-state batteries.

Page no.5

Furthermore, thermogravimetric analysis showed that the residual liquid content after EAP polymerization was less than 0.3 wt% (**Supplementary Fig. 18**), which was much lower than that of conventional drip-coating method, indicating the formation of a predominantly solid interfacial healing layer that firmly bridges the electrodes and OSE.

Thermogravimetric analysis (NETZSCH STA 449F5) was collected at 5.0 °C min⁻¹ between 25 °C and 800 °C under nitrogen atmosphere.

These are relatively minor points. I am pleased to recommend acceptance after minor revision.

Response:

We sincerely thank the Reviewer #1 for the positive evaluation and kind recommendation for acceptance of our revised manuscript. All minor points raised have been carefully addressed in the revised manuscript, We greatly appreciate the reviewer's constructive feedback, which has helped us further refine the manuscript, and we hope that the revised version meets with the reviewer's full approval.

Reviewer #2 (Remarks to the Author):

The authors have addressed my comments. I can recommend the publication of the manuscript.

Response:

Thanks for your kind work and consideration on the publication of our manuscript.

Reviewer #3 (Remarks to the Author):

The manuscript meets the publication requirements and is recommended for acceptance.

Response:

Thanks for your kind work and consideration on the publication of our manuscript.